# Synthesis, characterization and reactivity of a series of alkaline earth and rare earth iminophosphoranomethanide complexes
Matthew P. Stevens [1,6], Yu Liu[1,6], Elias Alexopoulos[1], Shifaa S. A. Xec Daudo[1], Rebecca R. Hawker[1], Adam Khan[1], Luis Lezama[2], Daniel Reta [3,4,5] ✉ & Fabrizio Ortu [1] ✉

There is a great demand from the synthetic community for readily accessible organometallic alkaline earth and rare earth compounds that can be used as synthons for the preparation of new complexes and materials. Here we report the use of the methanide ligand {CH(SiMe$_3$)P(Ph)$_2$ = NSiMe$_3$}$^-$ (NPC-H) in the stabilization of alkaline earth and rare earth organometallics and their potential use as synthetic precursors. *Bis-* and *tris*-methanide complexes were obtained following two methods: (1) deprotonation of the proligand with organometallic precursors; (2) salt elimination reactivity between potassium methanide and metal iodides. All compounds were characterised by multinuclear nuclear magnetic resonance, infra-red spectroscopy, elemental analysis, ultraviolet-visible spectroscopy and single crystal X-ray diffraction. Additionally, selected rare earth methanides were tested as synthetic precursors for the preparation of solvent-free rare earth complexes. Finally, the electronic structures of open-shell divalent rare earth methanides were probed using electron paramagnetic resonance, magnetometry studies and ab initio calculations.

Alkyl derivatives of alkaline earth (AE) and rare earth metals (RE) are highly desirable synthons of great utility in synthetic transformations and as organometallic starting materials[1–4]. The chemistry of AE and RE alkyl compounds is dominated by the high polarization of the M–C interaction, which makes the carbanion extremely nucleophilic[1,4,5]. This aspect, combined with the high electropositive character of the AE and RE metals, has a detrimental effect on the stability of alkyl derivatives, leading often to facile decomposition *via* inter- and intra-molecular C–H activation, α- and β-hydride elimination, solvent degradation and more[1,5]. To counter these drawbacks, several strategies can be employed, which include: (1) kinetic saturation of the metal coordination sphere; (2) chelation; (3) hyperconjugation to stabilize the methanide carbanion, {CR$_3$}$^-$[6]. Iminophosphoranomethanide ligands of the type {R–N = P(Ph)$_2$CH}$^-$ ($^R$BIPM-H, R = trimethylsilyl, *iso*-propyl, cyclohexyl, aryl) combine all of these aspects, and as a result have been used to stabilize metal complexes across the periodic table[7,8], including also alkaline earth (AE)[9–17] and rare earth (RE) methanides (Fig. 1)[9,18–39]. In the case of AE metals, both homoleptic (e.g., [Ba($^{TMS}$BIPM-H)$_2$][11]) and heteroleptic (e.g. [Ca($^{TMS}$BIPM-H) {N(SiMe$_3$)$_2$}][12,13] and [Ca($^{TMS}$BIPM-H)(BH$_4$)(THF)$_2$][10]) methanides have been reported. Moreover, numerous RE methanides have been stabilized

in the last two decades, comprising: 1) homoleptic *bis*-methanides (e.g. [Eu($^{TMS}$BIPM-H)$_2$])[37] and heteroleptic complexes of divalent REs (e.g. [{Eu($^{TMS}$BIPM-H)(THF)(μ-I)}$_2$])[9]; heteroleptic methanides of trivalent REs, stabilized with a variety of ancillary ligands, such as halides (e.g. [{RE($^{TMS}$BIPM-H)(Cl)(μ-Cl)}$_2$]; RE = Y, Sm, Dy, Er, Yb, Lu)[35], borohydrides (e.g. [Y($^{TMS}$BIPM-H)(BH$_4$)$_2$])[34], N-heterocyclic-carbenes (e.g. [RE($^{TMS}$BIPM-H)(I)$_2$(I$^{Me4}$)]; RE = Ce, Pr, Sm; I$^{Me4}$ = C(NMeCMe)$_2$)[20,27], and methanediides (e.g. [Dy($^{TMS}$BIPM)($^{TMS}$BIPM-H)])[40]. Some of these compounds have been used as initiators for lactide polymerization[10,18,38], and have also been employed as precursors towards the stabilization of *bis*-methanediide complexes[32,41]. Nonetheless, there are no reports of the use of these compounds as alkyl precursors in deprotonation reactions.

Le Floch and Mézailles developed an analogous ligand to BIPM-H, {(S = PPh$_2$)$_2$CH}$^-$[42], which was used in a similar fashion for the synthesis of homoleptic *bis*-methanide complexes for divalent metals or methanide/methanediide species (Fig. 1)[43,44]. Typical synthetic routes to these compounds include deprotonation of the proligand with Ln tribenzyl[45] or *tris*-amido precursors[46]. Alternatively, salt metathesis of a Group 1 salt of the ligand with a metal halide typically yields either halide-bridged[43] or monomeric heteroleptic species[28]. Moreover, Chen and co-workers

[1]School of Chemistry, University of Leicester, University Road, Leicester, UK. [2]Departamento de Química Orgánica e Inorgánica, Facultad de Ciencia y Tecnología, Universidad del País Vasco, B° Sarriena s/n, Leioa, Spain. [3]Faculty of Chemistry, The University of the Basque Country UPV/EHU, Donostia, Spain. [4]Donostia International Physics Center (DIPC), Donostia, Euskadi, Spain. [5]IKERBASQUE, Basque Foundation for Science, Bilbao, Euskadi, Spain. [6]These authors contributed equally: Matthew P. Stevens, Yu Liu. ✉e-mail: daniel.reta@ehu.eus; fabrizio.ortu@leicester.ac.uk

**Fig. 1 |** Bis-iminophosphoranomethanide and thiophosphoranomethanide ligands used in previous work with RE and AE metals, and iminophosphoranomethanide used in this work.

prepared an asymmetric analogue of the abovementioned thiophosphoranomethanide ligand, {(S = PPh$_2$)$_2$CH}$^−$, which incorporates silyl-substituents to replace one of the thiophosphorano arms, i.e., {S = P(Ph)$_2$CHSiR$_3$}$^−$ (R = Me, Ph, Fig. 1)[47]. However, this ligand has only been used to stabilize heteroleptic complexes supported by a tethered *bis*-diketiminate ligand, NacNac$^{NMe2}$ (NacNac$^{NMe2}$ = {N(Dipp)C(Me)CC(Me) N(CH$_2$CH$_2$NMe$_2$)}$^−$), such as [Sc(NacNac$^{NMe2}$){C(SiMe$_3$)PPh$_2$S-κ$^2$S, C}][48]. Similarly to BIPM-H, for both {(S = PPh$_2$)$_2$CH}$^−$ and {S = P(Ph)$_2$CHSiR$_3$}$^−$ there are no reports of their use as alkyl precursors for deprotonation reactions. All these ligands feature P(V) substituents on the central methanide, but there have been several reports that include the use of P(III) substituents and their borane adducts, a strategy used to prevent coordination of the lone pairs located on the P atoms with the metal centre[49–51].

There is a lot of interest in the AE and RE community for synthetic methodologies to obtain donor-free and low-coordination compounds, as these have physicochemical properties arising from their coordinative unsaturation[52–55]. Donor/solvent-free complexes are also ideal precursors for challenging reductive chemistry and isolation of low oxidation state species[56–60], all of which are usually incompatible with the use of ethereal solvents[61]. For this purpose, deprotonating reagents (e.g., alkyls, amides) are particularly attractive as they can react directly with protic proligands (e.g., amines, phenols) to give target complexes[39,62–65], with alkyl precursors providing the widest applicability owing to the high $p$Ka of their conjugated acid[1,2]. However, alongside some of the drawbacks described above, these are often obtained as ethereal solvent adducts or require the use of coordinating solvents, as for example: 1) AE and RE benzyls[39,63,66]; 2) α-silyl methanides, {CH$_2$(SiMe$_3$)}$^−$ – e.g. [Ca{CH$_2$(SiMe$_3$)}$_2$(THP)$_4$] (THP = tetrahydropyrano)[67], [RE{CH$_2$(SiMe$_3$)}$_2$(THF)$_3$] (RE = Y, Sm, Gd-Lu)[68–70].

We were therefore surprised by the fact that the use of the aforementioned methanides in deprotonation reactions had been completely overlooked. We reasoned that a potential deterrent towards the use of these chelating methanides could be identified in the high stability of some of these complexes and also, in the case of RE metals, the lack of homoleptic *tris*-methanides. Therefore, we were intrigued to explore new methanide systems that could incorporate some of the electronic and steric stabilization of BIPM-H, and the additional tunability of the asymmetric {S = P(Ph)$_2$CHSiR$_3$}$^−$ methanide, which could then be employed to stabilize solvent-free AE and RE reagents for deprotonation reactions – i.e. {CH(SiMe$_3$)P(Ph)$_2$ = NSiMe$_3$}$^−$ (NPC-H, Fig. 1). This ligand system was originally reported by Lappert and co-workers, together with the proligand Me$_3$SiN = P(Ph)$_2$CH$_2$SiMe$_3$ (NPC-H$_2$)[71]. The methanide NPC-H was then used to form complexes of tin(II) and lead(II)[72,73]; additionally, the group 1 salt [Li(NPC-H)] was reacted with benzonitrile to yield the insertion product [Li{N(SiMe$_3$)C(Ph)CHP(Ph)$_2$ = NSiMe$_3$}][74]. Recently, we have reported the use of NPC-H and the more sterically demanding $^{TBDMS}$NPC-H for the preparation of alkali metal complexes, including several separate ion

pair derivatives[6]. Despite the rich AE and RE methanide chemistry (*vide supra*), the NPC-H ligand has never been used with these metals. In this work, we develop further the chemistry of this methanide ligand and report on the preparation of AE and RE(II) *bis*-methanide complexes, and RE(III) *tris*-methanide complexes, together with their applications as deprotonation reagents.

## Results and Discussion
### Synthesis and reactivity
A deprotonation methodology was employed to prepare the homoleptic complexes [M(NPC-H)$_2$(THF)$_x$], **1-M** (M = Mg, Eu, Yb, x = 0; M = Ca, x = 0, 1; M = Sr, x = 0, 2; M = Ba, x = 2). In the case of Mg, commercially available $^n$Bu$_2$Mg was employed; for the remainder, dibenzyl precursors [M(CH$_2$Ph)$_2$(THF)$_x$] (M = Ca–Ba, Sm, Eu, Yb) were prepared and used in situ (Scheme 1)[63,66,75]. Mg, Ca and Yb derivatives were obtained as THF adducts as shown from the nuclear magnetic resonance (NMR) spectroscopic analysis of crude products, however, coordinated solvents could be removed through prolonged exposure to vacuum. Conversely, coordinated THF could not be removed from Sr and Ba derivatives upon exposure to vacuum or *via* recrystallization. Nonetheless, in one occasion **1-Sr** was obtained as a solvent-free compound upon crystallization, but we could not reproduce this method reliably (*vide infra*). Repeated attempts to isolate **1-Sm** were unsuccessful; however, crystals of [Sm(NPC-H)$_3$] (**2-Sm**) were instead obtained (Scheme 1). This indicates that the product had oxidised, possibly as part of the disproportionation of Sm(II) to Sm(III) and Sm(0), though reactivity with adventitious oxygen could not be ruled out. We found that a slight excess of metal dibenzyl precursors was required in the preparation of **1-M** to ensure no contamination of the product with proligand. This was likely due to the incomplete formation of the metal dibenzyl reagents generated in situ. We also found that it was not possible to purify the product if it was contaminated with proligand through recrystallization or *via* washings with hydrocarbons, particularly since the heavier **1-M** have appreciable solubility even in apolar solvent media. Despite these issues, the yields of **1-M** were consistently high (70-90%).

We also investigated salt metathesis as an alternative methodology for the synthesis of **1-M**, employing the ligand transfer reagent [{K(NPC-H)}$_2$] and MI$_2$(THF)$_x$ (M = Mg–Ba, x = 0; M = Eu, Yb, x = 2). This was successful in all trialled cases, with yields comparable to the deprotonation route, thus avoiding synthetic issues associated with the use of metal dibenzyl precursors (Scheme 1). When the method was extended to SmI$_2$(THF)$_2$, we were able to isolate the target Sm(II) *bis*-methanide complex [Sm(NPC-H)$_2$(THF)$_2$] (**1-Sm·(THF)$_2$**). We then decided to expand this methodology to other trivalent REs and targeted compounds of the general formula [RE(NPC-H)$_3$] (**2-RE**), choosing diamagnetic metals (Y, La) for reaction monitoring and a small selection of light REs (Pr and Sm) to further establish our methodology. Despite repeated recrystallizations and purification attempts, samples of **2-La** and **2-Pr** were always contaminated with

**Scheme 1 |** Synthesis of **1-M** and **2-Sm** from the reaction of NPC-H$_2$ with AE and Ln dibenzyl precursors. Synthesis of **1-M** and **2-RE** from salt metathesis reaction between [{K(NPC-H)}$_2$] and MI$_2$(THF)$_x$ (M = Mg–Ba, x = 0; M = Eu, Yb, x = 2) or [REI$_3$(THF)$_x$] (RE = Y, Sm, x = 3.5; RE = La, Pr, x = 4).

**Scheme 2 |** Reactivity of **1-Yb** and **2-RE** with HN(SiMe$_3$)$_2$ and 2,6-$^t$Bu$_2$C$_6$H$_3$OH.

various amounts of proligand, which prevented complete spectroscopic and analytical characterization.

RE and Ln organometallic reagents are often used as synthetic precursors in deprotonation reactions, with common reagents being alkyls, benzyls and α-silyl-alkyls[2]. Solvent-free organometallic precursors are relatively rare, and as such we were intrigued by the possibility of using **1-M** and **2-RE** as deprotonating agents in non-ethereal solvents to generate solvent-free complexes (Scheme 2). Therefore, we decided to test the reactivity of **1-M** and **2-RE** with 2,6-di-*tert*-butylphenol (2,6-$^t$Bu$_2$C$_6$H$_3$OH) and HN(SiMe$_3$)$_2$, respectively (Scheme 2), focusing on diamagnetic species (**1-Yb**, **2-Y** and **2-La**) amenable to screening *via* multinuclear NMR. The reactivity between **1-Yb** and amine or substituted phenol is straightforward,

leading to quantitative conversion at room temperature of **1-Yb** to [{Yb(OC$_6$H$_3$$^t$Bu$_2$-2,6)(μ-OC$_6$H$_3$$^t$Bu$_2$-2,6)}$_2$] and [{Yb[N(SiMe$_3$)$_2$][μ-N(SiMe$_3$)$_2$]$_2$}$_2$], respectively, as confirmed by the presence of the products in the $^1$H NMR spectra (Figs. S56, S57, S59 and S60), referenced against the characteristic resonances of [{Yb(OC$_6$H$_3$$^t$Bu$_2$-2,6)(μ-OC$_6$H$_3$$^t$Bu$_2$-2,6)}$_2$] and [{Yb[N(SiMe$_3$)$_2$][μ-N(SiMe$_3$)$_2$]$_2$}$_2$] previously reported in the literature[76]. Additionally, the formation of proligand NPC-H$_2$ is clearly evidenced in the $^{31}$P{$^1$H} NMR spectra (Figs. S58 and S61). Higher temperatures (80 °C) were required for the reactions between **2-RE** (RE = Y, La) and 2,6-di-*tert*-butylphenol. The formation of complexes [RE(OC$_6$H$_3$$^t$Bu$_2$-2,6)$_3$] (RE = Y, La) was confirmed by the presence of characteristic resonances in the $^1$H NMR spectra referenced against previous literature reports

**Fig. 2 |** $^1$H VT NMR study of **1-Mg** at 25–105 °C in d$_8$-toluene in the region ± 0.4 ppm.

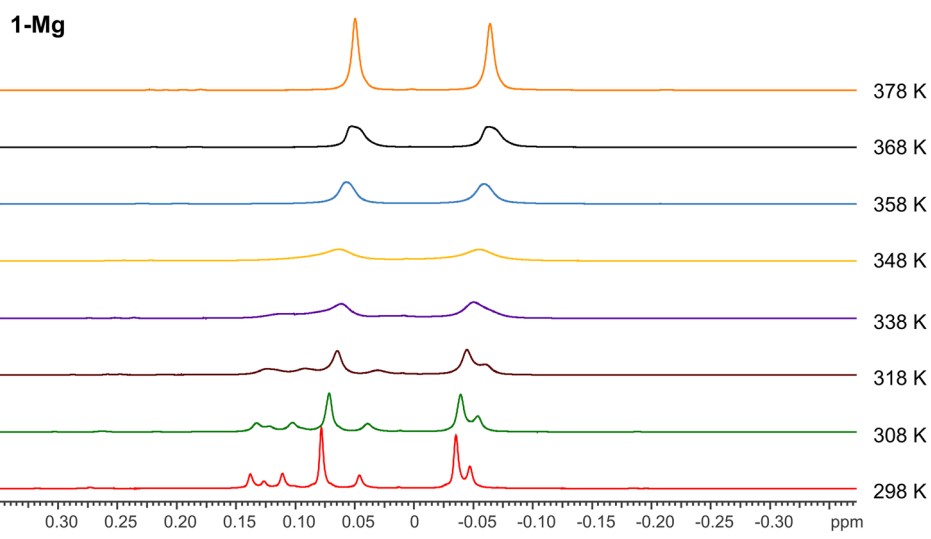

**1-Mg**

378 K
368 K
358 K
348 K
338 K
318 K
308 K
298 K

0.30  0.25  0.20  0.15  0.10  0.05  0  -0.05  -0.10  -0.15  -0.20  -0.25  -0.30  ppm

(Figures S64, S65, S70 and S71)[77,78]; additionally, the same peaks assigned to NPC-H$_2$ were observed in the $^{31}$P{$^1$H} NMR spectra in both cases. It should be noted that additional amine or phenol signals were detected in the $^1$H NMR spectra, which could be due to the presence of either unreacted material or incomplete conversion to *mono-* or *bis-*methanide products, as evidence also by the presence of additional resonances which we could not assign (Figs. S64 and S70). Finally, reactivity of **1-Y** and **1-La** did not produce the target [RE{N(SiMe$_3$)$_2$}$_3$] complexes, even under forcing conditions (80 °C for several days). To confirm this, owing to the close proximity of key resonances in the $^1$H NMR spectra, we used isolated [Y{N(SiMe$_3$)$_2$}$_3$] and [La{N(SiMe$_3$)$_2$}$_3$] as references for spectroscopic analyses (Figs. S62 and S67). For these reactions, it should be noted that proligand NPC-H$_2$ is still produced, thus suggesting that deprotonation of HN(SiMe$_3$)$_2$ is taking place; it is therefore possible that heteroleptic complexes are formed (e.g., [RE(NPC-H)$_2${N(SiMe3)$_2$}] or [RE(NPC-H){N(SiMe3)$_2$}$_2$]) rather than the target *tris*-amide, but these could not be unequivocally identified.

## NMR spectroscopy

The $^1$H NMR spectrum of **1-Mg** contains a complex peak arrangement (Fig. S5) corresponding to the methanide CH bound directly to the magnesium atom, which resolves into four singlets ($\delta_H$ 1.24, 1.19, 1.14 and 1.06 ppm) in the $^1$H{$^{31}$P} spectrum (Figs. S8–10). We carried out a variable temperature $^1$H NMR (VT-NMR) study in toluene-d$_8$ in order to ascertain whether these multiple signals were due to the presence of multiple conformations, rather than impurities. At 105 °C, these signals coalesce into a single broad doublet ($\delta_H$ 1.02 ppm, $^2J_{HP}$ = 14.7 Hz), thus suggesting that all signals belong to the same species (**1-Mg**), with restricted rotations or locked conformations overcome in the high temperature regime (Fig. S6 and S17–19). Similar behaviour is seen in the region between +0.5 and –0.5 ppm, where the two different SiMe$_3$ groups resonate (Fig. 2). Further evidence of non-interconvertible conformations and impeded rotations being present at room temperature is also seen in the $^{31}$P{$^1$H}, $^{29}$Si{$^1$H}, and $^{13}$C{$^1$H} NMR spectra (see SI).

The $^1$H NMR spectra of all the other diamagnetic complexes **1-M** (M = Ca–Ba, Yb) are much simpler by comparison. Notably, as Group 2 is descended, the $^1$H NMR chemical shift of the C–H methanide moves steadily upfield with more electropositive AE metals, to the point with **1-Ba·(THF)$_2$** that it lies upfield of the SiMe$_3$ resonance signals (0.08 ppm *vs.* 0.15 for **1-Sr·(THF)$_2$** and 0.18 ppm for **1-Ca**). Similarly to what is observed with the $^{13}$C{$^1$H} NMR resonances, the $^{29}$Si{$^1$H} NMR spectra of **1-M** do not vary a great deal between metals. NPC-H$_2$ has very different resonances to **1-M**, and **1-Mg** is an outlier in that it does not conform to the consistent chemical shifts detected for the heavier congeners. The spectra exhibit two signals

**Table 1 | $^1$H and $^{13}$C NMR chemical shifts of methanide carbon of 1-M (M = Mg–Ba, Yb), together with $^{31}$P{$^1$H} NMR chemical shifts**

| Compound | $\delta_P$ | $\delta_H$ of M–CH ($^2J_{HP}$ / Hz) | $\delta_C$ of M–CH ($^1J_{CP}$ / Hz) |
|---|---|---|---|
| **NPC-H$_2$** | –0.40 | 1.59 (15.2) | 19.9 (69.3) |
| **1-Mg** | 30.1* | 1.05* (14.7)† | 17.8 (59.4) |
| **1-Ca** | 22.4 | 0.43 (14.3) | 19.3 (70.8) |
| **1-Sr·(THF)$_2$** | 22.3 | 0.30 (14.3) | 20.8 (79.5) |
| **1-Ba·(THF)$_2$** | 20.3 | 0.08 (11.7) | 21.6 (87.7) |
| **1-Yb** | 19.7 | 0.77 (12.6) | 24.2 (88.4) |

*Centre of grouping at 25 °C; †$\delta_H$ at 105 °C is 1.06 ppm, coupling constant extracted at 105 °C.

which were assigned based on their differing $^2J_{SiP}$ coupling constants (Table S1) – the larger splitting was assigned to Si(a) and the smaller to Si(b) (Fig. S55). X-ray diffraction studies further support this assignment: in the solid state structures the N = P double bond are shorter than the C–P single bond (*vide infra*, Structural Characterisation), thereby increasing the coupling strength to phosphorus as the greater spatial overlap between the respective *s*-orbitals would lead to better communication of nuclear spin between nuclei.

$^{31}$P{$^1$H} NMR spectra were also found to be extremely diagnostic of the formation of **1-M**. The free proligand exhibits a single resonance at –0.40 ppm, while deprotonation and coordination of AE metals or Yb to the methanide carbon changes the resonance frequency to 20–30 ppm (Table 1). The greatest difference was observed for **1-Mg** ($\delta_P$ = 30.1 ppm, centre of grouping). The heavier analogues of **1-M** (M = Ca, Sr, Ba, Yb) consistently exhibit $^{31}$P NMR resonances around 20 ppm. The complementary $^2J_{PH}$ coupling could not be observed in the $^{31}$P NMR spectrum (Fig. S13b). In the $^{31}$P{$^1$H} NMR of **1-Yb**, $^{171}$Yb satellites could be observed ($^{171}$Yb, I = ½, 14.3% abundance), with a $^2J_{PYb}$ coupling constant of 90–95 Hz (Figure S34, inset). Yb–P coupling is mainly reported for $^1J_{PYb}$, which is much larger in magnitude (500–900 Hz)[79,80]. Reports of longer range couplings are scarcer and of lower magnitude (e.g. [Yb(C$_4$PMe$_4$)$_2$($\kappa^2$N, N-2,6-{CH$_2$N = PPh$_3$}$_2$C$_5$H$_3$N)] $^2J_{PYb}$ = 172 Hz; [({C$_6$H$_{11}$}$_2$PC$_2$H$_4$P{C$_6$H$_{11}$}$_2$)Pt($\mu$-H)$_2$Yb(C$_5$Me$_5$)$_2$] $^3J_{PYb}$ = 93 Hz)[80,81]. The $^{13}$C{$^1$H} NMR shows an interesting trend down Group 2, with regards to the M–CH resonance, with chemical shifts increasing steadily moving from Mg to Ba. This is analogous to the trend noted earlier in the $^{31}$P{$^1$H} NMR spectra of **1-M**. Further to this, the $^1J_{CP}$ coupling constant also steadily increases down the group from 59.4 Hz (**1-Mg**) to 87.7 Hz (**1-Ba·(THF)$_2$**) indicating a stronger magnetic communication between the two atoms.

The $^{171}$Yb{$^1$H} NMR spectrum of **1-Yb** (Figure S36) contains a single broad resonance at 1046.5 ppm (FWHM = 290 Hz). This signal is quite downfield relative to many reported ytterbium alkyls, but still within typical regions (600–1200 ppm)[82–85]. It should be noted that Yb(II) Cp* complexes (e.g., [Yb(Cp*)$_2$(THF)$_2$] and [Yb(Cp*)$_2$(OEt)$_2$]) exhibit signals in their $^{171}$Yb NMR spectra in the region between 30-90 ppm[86]. There are no reported $^{171}$Yb NMR chemical shifts for ytterbium complexes with iminophosphoranomethanide ligands, however there are ample examples with other ligand systems. Lappert and co-workers reported $^{171}$Yb NMR data for a wide series of Yb amides and aryloxides, with chemical shifts ranging between 230-1300 ppm[76,86]. Additionally, Junk and co-workers reported $^{171}$Yb NMR data for several Yb formamidinate complexes (e.g., [Yb(Ar-Form)I(THF)$_3$], Form = ArNCHNAr, Ar = 2,6-Me$_2$C$_6$H$_3$, 2,6-$^i$Pr$_2$C$_6$H$_3$; $\delta_{Yb}$ = 552 ppm)[87]. These typically exhibited resonances in their $^{171}$Yb NMR spectra around 400–600 ppm, although it should be noted that these also contained a halide atom additionally ligating the metal atom. Moreover, Lappert and co-workers reported several Yb homoleptic and heteroleptic β-diketiminate complexes[88,89]. The homoleptic species ([Yb(L)$_2$], L = N(SiMe$_3$)C(Ar)C(H)C(Ar)N(SiMe$_3$), Ar = Ph, Tol, C$_6$H$_4$Ph-4) gave characteristic resonances in the region 2650 ± 200 ppm[89], while heteroleptic complexes ([Yb(L)(R)], R = I, N(SiMe$_3$)$_2$, CH(SiMe$_3$)$_2$) resonated closer to 1000 ppm[88]. It would therefore appear that the NPC-H ligand is quite shielding if compared with the more familiar β-diketiminate ligands while it appears on a par with Junk's formamidinate systems. Réant et al. isolated a series of s- and f-block metal silanides to investigate bond covalency via $^{29}$Si NMR, and reported a $^{171}$Yb NMR chemical shift of [Yb(Si$^t$Bu$_3$)$_2$(THF)$_2$] of 1044.64 ppm[90]. This is quite close to the chemical shift of **1-Yb**, and also lies within the typical range of ytterbium alkyls.

The $^1$H NMR spectra of both **1-Sm·(THF)$_2$** and **1-Eu** present significant broadening that precludes a reasonable integration (Figures S37, S39 and S40). No signals were observed in the $^{13}$C{$^1$H} spectra, however a distinct broad peak was observed in the $^{31}$P{$^1$H} NMR spectra of **1-Eu**, resonating at 35.3 ppm ($\nu_{\frac{1}{2}}$ = 270 Hz); this is strikingly different from the $^{31}$P chemical shift reported by Roesky and Wiecko for [Eu(BIPM-H)$_2$], which was measured at 532 ppm[37]. **1-Eu** was probed for its magnetic susceptibility by the Evans NMR method in benzene, and a χT of 6.57 cm$^3$ mol$^{-1}$ ($\mu_{eff}$ = 7.25 $\mu_B$) was calculated[91]. This is lower than theoretical value for a spin-only ($S$ = 7/2) ground state, and also lower than the experimental value for other Eu(II) complexes (e.g., [Eu(Cp′)$_3$]$^-$ - Cp′ = C$_5$H$_4$SiMe$_3$) obtained by Meihaus et al. (7.31 cm$^3$ K mol$^{-1}$; $\mu_{eff}$ = 7.65 $\mu_B$)[92]. With the same method, we also measured the magnetic susceptibilities for **1-Sm** (3.58 $\mu_B$), **2-Sm** (2.98 $\mu_B$) and **2-Pr** (3.19 $\mu_B$). The $\mu_{eff}$ value of **1-Sm** is in line with previously reported Sm(II) complexes, which all deviate from a $^7F_0$ diamagnetic ground state because of mixing with low-lying excited states[93–95]. In the case of **2-Sm**, $\mu_{eff}$ is higher than the calculated value for the free ion (Sm$^{3+}$ 0.85 $\mu_B$)[39], though it is not uncommon for Sm(III) complexes to have higher magnetic moments owing to the presence of low-lying excited states that mix with the $^6H_{5/2}$ ground state[96]. The $\mu_{eff}$ value for **2-Pr** is slightly lower than predicted theoretical values (3.58 $\mu_B$)[39], but this could be due to the presence of diamagnetic impurities in the sample, which could not be removed upon recrystallization (*vide supra*).

As mentioned above, NMR analysis of various samples of **2-La** revealed the presence of residual proligand, NPC-H$_2$, as a persistent impurity, likely obtained as the result of partial decomposition. Despite these impurities, the $^{31}$P{$^1$H} NMR spectrum clearly shows resonances corresponding to **2-La** at 27.8 ppm. In the case of **2-Y**, the same restricted rotation as for **1-Mg** was observed in its $^1$H spectrum (Fig. S41), $^{31}$P{$^1$H} (Fig. S43) and $^{29}$Si{$^1$H} (Fig. S45) NMR spectra. The $^{31}$P{$^1$H} spectrum of **2-Y** shows four distinct peaks resonating at 29.9, 31.7, 32.2, and 33.6 ppm respectively, integrating at ratios of 0.1:0.1:0.7:0.1. Therefore, a VT-NMR study was undertaken (Fig. S44) which showed coalescence of the four signals into one at approximately 85 °C at 32.5 ppm, with new signals growing in at 24.4 ppm and 25.7 ppm at approximately 75 °C. These most likely correspond to different conformers of **2-Y** that are accessible at these higher temperatures. An analogous study was undertaken with **2-La**, albeit descending in

## Table 2 | Selected bond lengths (Å) of 1-M (M = Mg–Ba, Sm, Eu, Yb)

| Complex | M–CH | M–N |
|---|---|---|
| **1-Mg**\* | 2.189(6)–2.244(4) | 2.069(3)–2.118(3) |
| **1-Ca** | 2.498(12)–2.549(4) | 2.337(4)–2.372(7) |
| **1-Ca · (THF)** | 2.6198(14), 2.6137(14) | 2.3894(12), 2.3961(12) |
| **1-Sr · (THF)$_2$** | 2.804(4), 2.851(4) | 2.593(3), 2.636(4) |
| **1-Ba · (THF)$_2$** | 2.955(4), 3.026(3) | 2.769(3), 2.729(3) |
| **1-Sm · (THF)$_2$** | 2.783(5), 2.823(6) | 2.599(4), 2.641(4) |
| **1-Eu** | 2.636(10)–2.687(6) | 2.504(5)–2.555(10) |
| **1-Yb** | 2.43(2)-2.575(5) | 2.337(5)-2.37(2) |

\*distances obtained from two different polymorphs.

temperature (Fig. S49). It can also be seen from these studies that $^2J_{PY}$ coupling is visible (8.2 Hz). $^1J_{PY}$ scalar coupling constants are typically around 40–50 Hz for neutral phosphine ligands[97–99], whereas in yttrium BIPM complexes, the $^2J_{PY}$ coupling constants are more in the region of 12–15 Hz[19,22,100]. From this, it can be seen that the $^2J_{PY}$ constant in **2-Y** is relatively small, although plausible. In the $^{13}$C{$^1$H} NMR, the methanide carbon signal resonates as a doublet of doublets, with a $^1J_{CP}$ coupling constant of 47 Hz (*c.f.* Table 2), while the $^1J_{CY}$ coupling constant is much smaller at 8.6 Hz. In yttrium BIPM complexes, the $^1J_{CY}$ coupling constants are typically in the region of 4–8 Hz, thus slightly lower than the analogous constant observed for **2-Y**[22]. The $^1$H NMR spectrum of **2-Sm** is clearly affected by paramagnetism, with signals spread between –13 and +11 ppm. The $^{31}$P{$^1$H} NMR spectrum of 2-Sm is very low intensity, but shows two broad resonances at 68.0 ppm (FWHM = 169 Hz) and 73.3 ppm (FWHM = 186 Hz). This is likely indicative of the presence of non-interconvertible conformations and impeded rotations as seen in **1-Mg** and **2-Y**. These data together show that **2-RE** have a similar non-interconvertible conformation and impeded rotation phenomenon that is seen in **1-Mg**. It is noteworthy that the greater size of the lanthanide ions relative to magnesium[101] is more than compensated for by the greater filling of the coordination sphere by the three NPC-H ligands.

## X-ray crystallography

Single crystals suitable for XRD studies were obtained for all **1-M** complexes, which confirmed the proposed connectivity in all cases. The crystal structure of **1-Mg** (Fig. 3) is reminiscent of the magnesium *bis*(iminophosphorano) methanide complexes [Mg($^{TMS}$BIPM-H)(Cl)]$_2$ and [Mg($^{TMS}$BIPM-H)(I)] reported by Wei and Stephan[102]. In these species the Mg–C bond distances [Cl: 2.461(8) Å; I: 2.573(6) and 2.639(7) Å] are considerably longer than in **1-Mg** [2.189(6)–2.244(4) Å] (Table 2), which is likely due to the geometry of the BIPM-H ligand holding the two atoms further apart. Furthermore, the Mg–N bond distances in the BIPM-H complexes [Cl: 2.104(7) and 2.164(7) Å; I: 2.088(5)–2.121(6) Å] are within the range of the bond lengths in **1-Mg** [2.069(3)–2.118(3) Å]. Two polymorphs of **1-Mg** were obtained, crystallizing in monoclinic $P2_1/c$ and $C2/c$, respectively; in both cases, the coordination geometry of the Mg centre is distorted tetrahedral, with subtle variations across the two polymorphs ($P2_1/c$: $\tau_4$ = 0.67; $C2/c$: $\tau_4$ = 0.53).

Crystal structures were also obtained for **1-Ca** and **1-Sr** (Fig. 3 and S84), and their corresponding coordinated THF adducts – **1-Ca·THF** and **1-Sr·(THF)$_2$** (Fig. 4). However, in the case of unsolvated **1-Sr**, the data quality was quite poor, so whilst the connectivity is clear-cut, it is not possible to draw statistically meaningful conclusions regarding bond distances and angles. The structures of **1-Ca** and **1-Sr** display a distorted tetrahedral coordination geometry (**1-Ca**: $\tau_4$ = 0.54). **1-Ca** crystallized in the $P$-1 space group, while **1-Sr** crystallized in the monoclinic space group $P2_1/c$. Compared to the homoleptic *bis*(iminophosphorano)methanide complex [Ca($^{TMS}$BIPM-H)$_2$] [Ca–N: 2.4443(12)–2.6786(8) Å; Ca–C 2.7241(13) Å and 2.8048(12) Å][15], the Ca–N bond distances in **1-Ca** and **1-Ca·THF** are

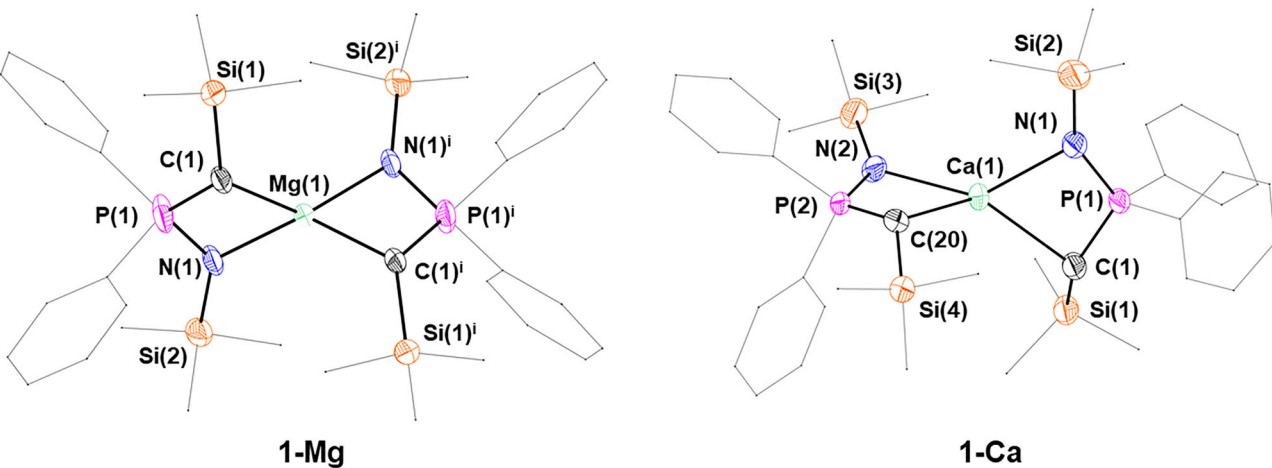

**Fig. 3 | X-ray crystal structures of 1-Mg and 1-Ca with selected labelling.** Ellipsoids set at 50% probability level; hydrogen atoms and disorder are excluded, and phenyl groups and trimethylsilyl groups are shown as wireframe for clarity. Symmetry operation used to generate equivalent atoms (**1-Mg**): i = $-x, +y, \frac{3}{2} - z$.

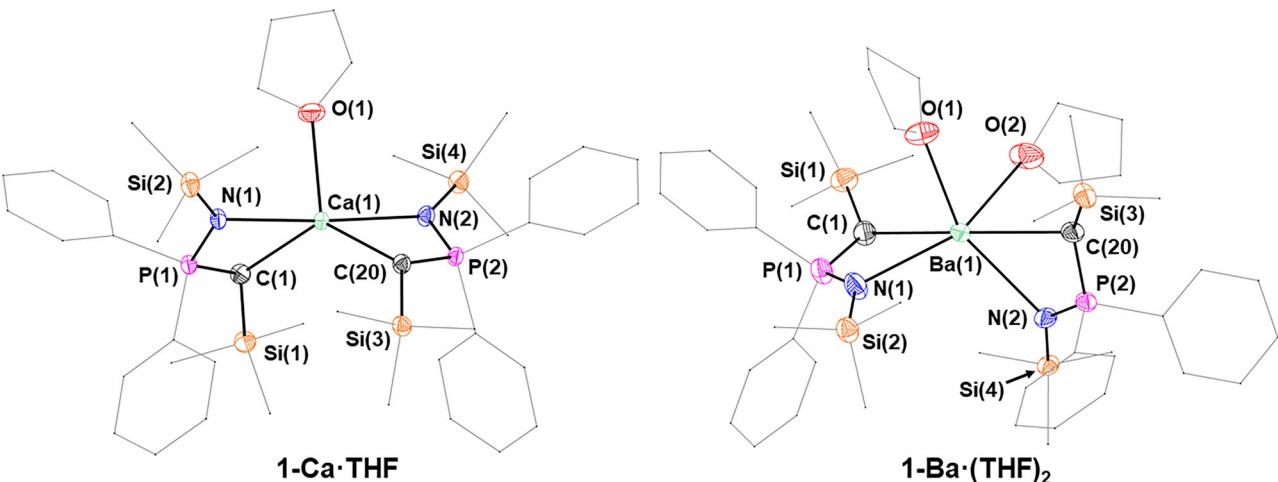

**Fig. 4 | X-ray crystal structures of 1-Ca·THF and 1-Ba·(THF)$_2$ with selected labelling.** Ellipsoids set at 50% probability level; hydrogen atoms and disorder are excluded and THF molecules, phenyl groups and trimethylsilyl groups are shown as wireframe for clarity.

shorter [2.337(3)–2.3961(11) Å], as are the Ca–C distances [2.498(12)–2.6198(14) Å]. This is likely due to the larger steric bulk of the BIPM-H ligand preventing closer contact between the metal and the coordinating atoms. Roesky and co-workers also prepared a calcium BIPM-H complex, [Ca($^{TMS}$BIPM-H)(BH$_4$)(THF)$_2$], which had less steric clash[10]. Hence, the bond lengths [Ca–C 2.720(4) Å; Ca–N 2.488(3) and 2.493(4) Å] are a little shorter compared to Harder's [Ca($^{TMS}$BIPM-H)$_2$][15]. **1-Sr·(THF)$_2$** (Fig. 5) and **1-Ba·(THF)$_2$** (Fig. 4) were also obtained, and these both crystallized in the $P2_1/n$ space group. Roesky and co-workers prepared isostructural strontium and europium borohydride complexes, [M($^{TMS}$BIPM-H)(BH$_4$)(THF)$_2$] (M = Sr, Eu); the Sr analogue exhibits similar bond lengths [Sr–C: 2.879(3) Å; Sr–N: 2.621(3) Å and 2.624(2) Å] to **1-Sr·(THF)$_2$** [Sr–C: 2.851(4) Å and 2.804(4) Å; Sr–N: 2.593(4) Å and 2.636(3) Å][14]. Additionally, Harder and co-workers obtained [Ba($^{TMS}$BIPM-H)$_2$], which has a marginally longer Ba–C bond [3.112(2) Å] than **1-Ba·(THF)$_2$** [2.955(4) Å and 3.026(3) Å][11]. Conversely Ba–N bond lengths in [Ba($^{TMS}$BIPM-H)$_2$] [2.726(2) Å and 2.902(2) Å][11] and **1-Ba·(THF)$_2$** [2.729(3) Å and 2.769(3) Å] are similar.

**1-Sm·(THF)$_2$** (Fig. 5) crystallizes in the monoclinic space group $P2_1/n$ and exhibits M–C [2.783(5) Å and 2.823(6) Å] and M–N distances [2.599(4) Å and 2.641(4) Å] that are comparable with those of **1-Sr·(THF)$_2$** (Table 2). There is only one example of structurally authenticated, homoleptic Sm(II) *bis*(iminophosporano)methanide reported in the literature,

[Sm($^{Mes}$BIPM)$_2$][25], which exhibits similar Sm–N [2.592-2.608 Å] distances to **1-Sm·(THF)$_2$**, whilst the and Sm–C bond distances [2.877(5) Å and 2.900(5) Å] are slightly elongated. Additionally, few other 'supported' homoleptic dialkyl Sm(II) complexes have been reported, most notably the complex [Sm{C[Me$_2$P(BH$_3$)](Me$_3$Si)$_2$}$_2$(THF)] isolated by Izod and co-workers[50]. In the case of [Sm{C[Me$_2$P(BH$_3$)](Me$_3$Si)$_2$}$_2$(THF)], only one of the Sm–C distances [2.827(9) Å and 2.853(8) Å] is slightly elongated compared to **1-Sm·(THF)$_2$**. **1-Sm·(THF)$_2$** is isostructural with **1-Sr·(THF)$_2$**, displaying statistically equivalent M–N and M–C bond distances (Table 2).

**1-Yb** (Fig. 6) is isostructural with **1-Ca**, displaying the same distorted tetrahedral coordination geometry ($\tau_4 = 0.53$). Roesky and co-workers synthesized the closely related ytterbium methanide [Yb($^{TMS}$BIPM-H)$_2$][103], and it is worth noting also the heteroleptic methanide complexes [Yb($^{TMS}$BIPM-H)(BH$_4$)(THF)$_2$] and [Yb[S = P(Ph)$_2$C(H)P(Ph)$_2$ = NSiMe$_3$](BH$_4$)(THF)$_2$] reported by the same authors[23,104]. The Yb–C bond distances in [Yb($^{TMS}$BIPM-H)$_2$] [2.935(7) Å] are significantly longer than in **1-Yb** [2.43(2)–2.575(6) Å], as are the Yb–N distances [BIPM: 2.546(5) and 2.450(4) Å vs. 2.337(4)–2.37(2) Å]. Finally, **1-Eu** crystallized in the $P2_1/c$ space group (Fig. 6), without coordinated THF and displaying a distorted tetrahedral geometry ($\tau_4 = 0.50$). Roesky and co-workers have reported several Eu(II) methanide complexes, such as [Eu($^{TMS}$BIPM-H)(THF)$_2$(BH$_4$)][23], [Eu($^{TMS}$BIPM-H)(THF)$_3$][BPh$_4$] and [Eu($^{TMS}$BIPM-H)$_2$][37]. The latter is very closely related to **1-Eu**, displaying Eu–C bond

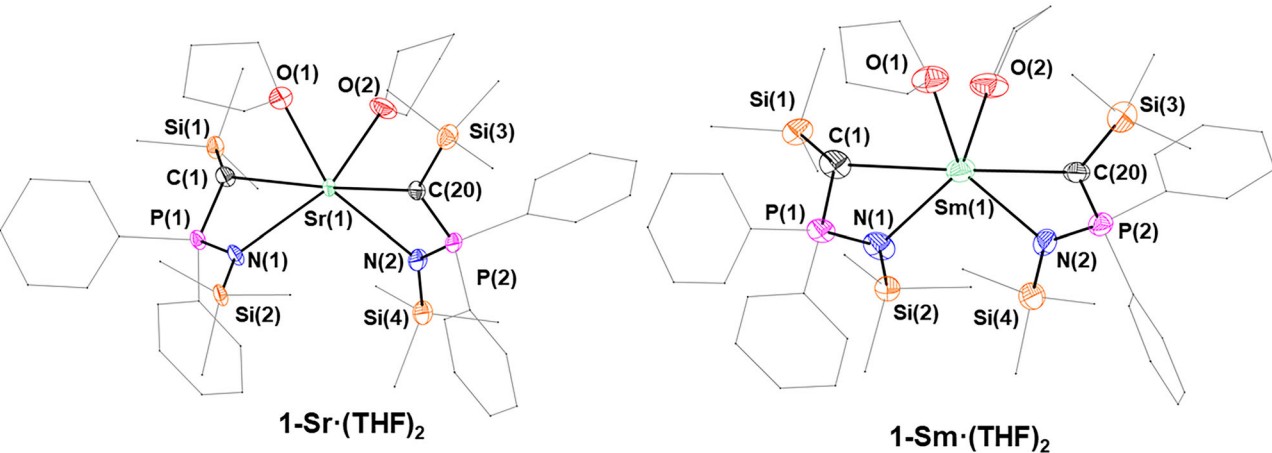

**Fig. 5 | X-ray crystal structures of 1-Sr·(THF)₂ and 1-Sm·(THF)₂ with selected labelling.** Ellipsoids set at 50% probability level; hydrogen atoms and disorder are excluded and THF molecules, phenyl groups and trimethylsilyl groups are shown as wireframe for clarity.

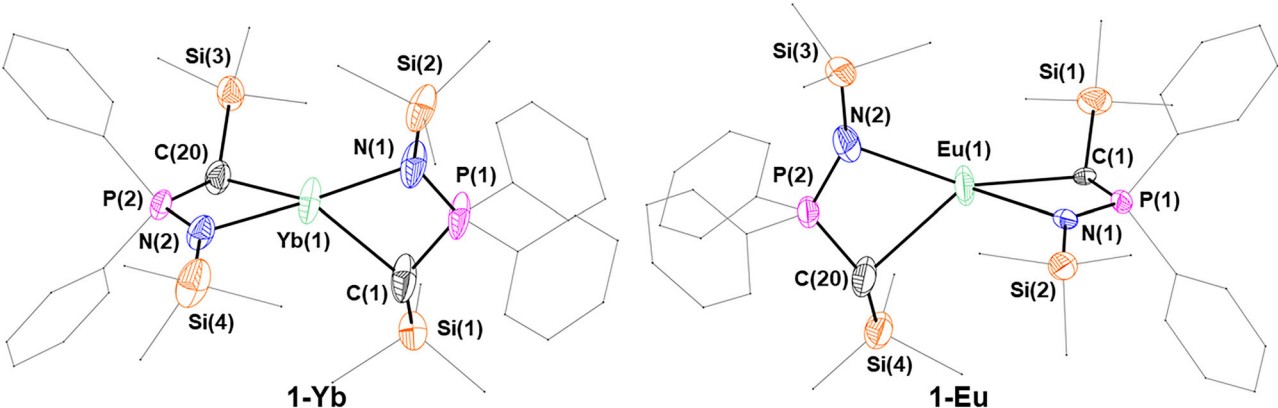

**Fig. 6 | X-ray crystal structures of 1-Eu and 1-Yb with selected labelling.** Ellipsoids set at 50% probability level; hydrogen atoms and disorder are excluded, and THF molecules, phenyl groups and trimethylsilyl groups are shown as wireframe for clarity.

distances [2.865(5) Å and 2.880(5) Å] that are significantly longer than the analogous bonds in **1-Eu** [2.636(10)–2.687(6) Å]. The reason for this is not clear, although crystal-packing effects may be responsible. Much as observed before, the Eu–N bond lengths in [Eu(^{TMS}BIPM-H)₂] [2.602(4)–2.778(4) Å] are also longer than in **1-Eu** [2.504(5)–2.555(10) Å]; this is likely due to the different coordination mode of the NPC-H ligand (i.e. bidentate, N^C) relative to BIPM-H (i.e. tridentate, N^C^N).

**2-Y**, **2-Pr** and **2-Sm** crystallize in the *P*-1 space group with very similar unit cell parameters, each featuring two independent molecules in the unit cell. **2-La** instead crystallizes in the hexagonal *P*6₃22 space group, with the three NPC-H ligands symmetrically related (Fig. 7) – the two donor atoms (methanide C and imide N) are completely equivalent in the model, exemplified by the asymmetric unit featuring only half of the NPC-H ligand. All four complexes adopt a twisted trigonal prismatic coordination environment about the metal, rather than the more familiar 6-coordinate octahedral shape. This is due to the small bite angle (ca. 65°) of the NPC-H ligand. In each complex, the three phosphorus atoms and lanthanide atom all lie in the same plane (average of sum of P–RE–P angles: **2-Y** 359.81°; **2-La** 360.00°; **2-Pr** 356.98°; **2-Sm** 359.89°). All reported ^RBIPM complexes consist of two ligands coordinating the metal centre because BIPM is too sterically demanding for three ligands to bind a single metal centre.

The closest reported analogue to **2-RE** comprises an iminophosphoranylaryl ligand, [C₆H₄P(Ph)₂ = NⁿBu]⁻. This forms homoleptic complexes of type [RE{C₆H₄P(Ph)₂ = NⁿBu}₃] (RE = Y, La, Nd)[105]. In [La{C₆H₄P(Ph)₂ = NⁿBu}₃] the La–N bond lengths [2.554(2)–2.632(3) Å] are shorter than in **2-RE** (Table 3). This may be due to less steric crowding

about the binding atoms (C, N). Unfortunately, no crystal structure of [Y{C₆H₄P(Ph)₂ = NⁿBu}₃] was reported, and there are no other analogous coordination environments with closely related ligands bound to early lanthanides. Alternatives are limited to lutetium, which is too small relative to even yttrium to permit valid comparisons[106,107]. Comparisons could also be drawn with the P(III)-stabilized methanides [RE{CH(PPh₂)₂}] (RE = La[108], Sm)[109], though in these cases the coordination of the {CH(PPh₂)₂}⁻ ligand is comparable to a classic η³-allyl donor and the RE–C distances [La–C 2.790(4)–2.908(5) Å; Sm–C = = 2.720(9)-2.787(9) Å] are elongated compared to **1-La** and **1-Sm**.

### UV-vis spectroscopy and magnetic studies

UV-visible absorption spectra were obtained for all **1-M·(THF)ₓ** complexes (Fig. 8). In the case of **1-Mg**, the spectrum was essentially featureless, besides an absorption around 283 nm. Descending Group 2, we also observed the appearance of two new absorption bands, which are clearest in the case of **1-Ca** (341 nm, ε = 2460 M⁻¹ cm⁻¹ and 425 nm, ε = 650 M⁻¹ cm⁻¹). These are substantially weaker than the absorption at 283 nm, and likely correspond to ligand-to-metal charge transfer (LMCT) transitions in the M–C and M–N bonds. This would also explain why the $\lambda_{max}$ decreases down the group – a more ionic bond would lower the energy of the transition, hence a lower $\lambda_{max}$. The three divalent lanthanide complexes (**1-Sm·(THF)₂**, **1-Eu**, and **1-Yb**) also showed the same transition ca. 280 nm, and the LMCT bands in the visible region. For **1-Eu** no $f \rightarrow f$ or $f \rightarrow d$ transitions could be distinguished. The spectrum of **1-Sm·(THF)₂** shows two features at 400 nm (ε = 525 M⁻¹ cm⁻¹) and 590 nm (ε = 335 M⁻¹ cm⁻¹) which have been

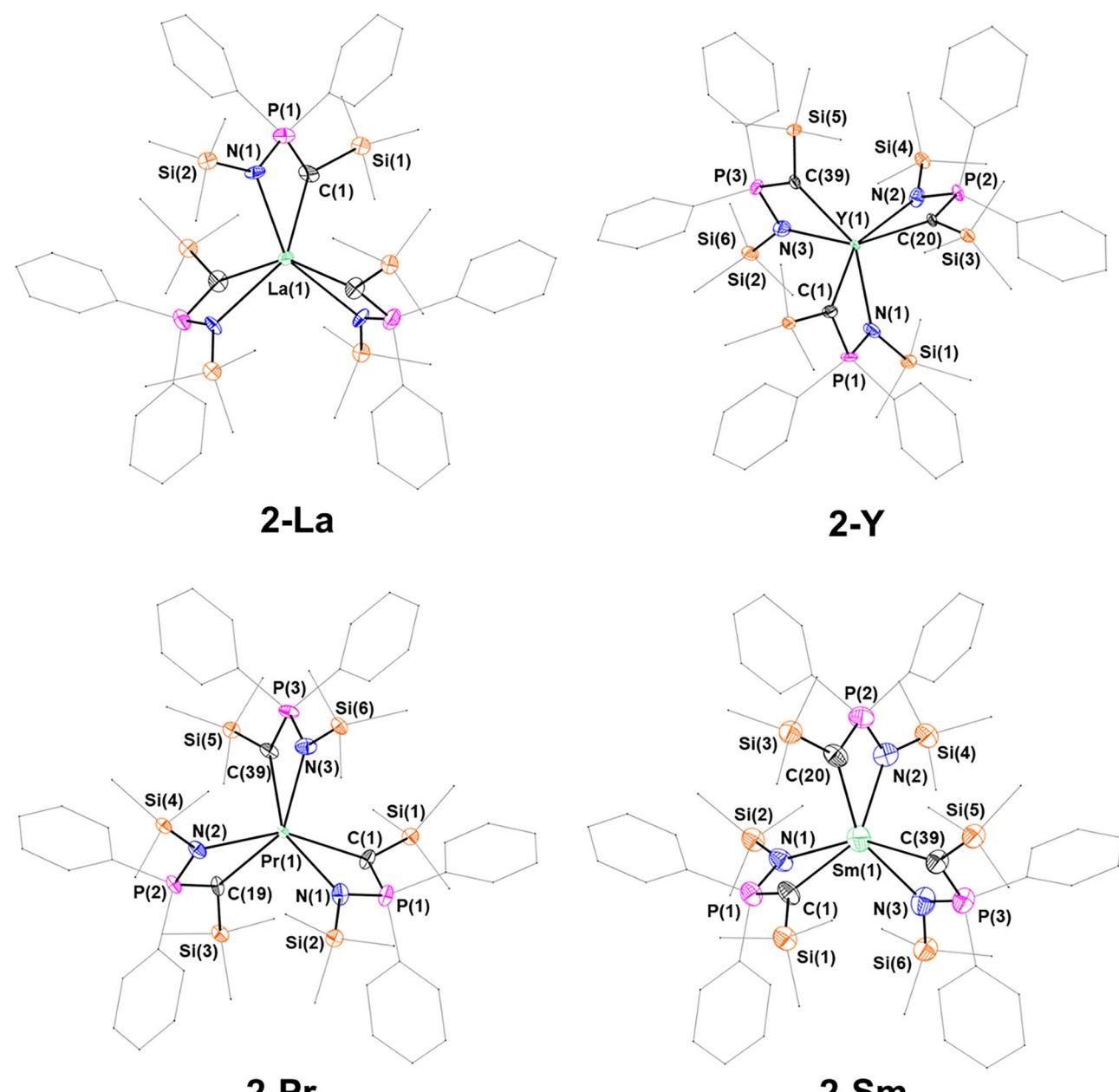

**Fig. 7 | Crystal structures of 2-Y, 2-La, 2-Pr and 2-Sm.** Ellipsoids set at 50% probability level; hydrogen atoms and disorder are excluded and phenyl groups and trimethylsilyl groups are shown as wireframe for clarity. Symmetry operations to generate equivalent atoms (**2-La**): i = −y + x, +x, ½ + z; ii = −y, + x−y, z; iii = −x, −y, ½ + z; iv = + y−x, − x, +z; v = + y, −x + y, ½ + z; vi = −y + x, −y, − z; vi = −x, −x + y, −z; vii = +y, +x, −z; viii = −y, −x, ½ −z; ix = + y−x, + y, ½−z; x = +x, +x−y, ½−z.

## Table 3 | Selected bond length (Å) of 2-RE

| Complex | M–CH | M–N |
|---------|------|-----|
| 2-Y | 2.519(6)–2.594(6) | 2.427(5)–2.484(6) |
| 2-La | 2.53(5) | 2.72(4) |
| 2-Pr* | 2.717(4)–2.794(4) | 2.436(4)–2.487(4) |
| 2-Sm | 2.426 (6)–2.739(6) | 2.435(6)–2.744(6) |

*distances obtained from primary disorder component.

previously observed in other Sm(II) complexes and assigned as $f \rightarrow d$ transitions[94,95]. In the case of **1-Yb**, a more concentrated sample (1.0 mM in toluene) revealed a weak absorption band at 400 nm ($\varepsilon = 110$ M$^{-1}$ cm$^{-1}$) which could be assigned as an $f \rightarrow d$ transition. UV-visible absorption spectra were also recorded for **2-Y**, **2-La**, **2-Pr** and **2-Sm** (see SI). In all cases, the spectra were essentially featureless, with **2-Sm** and **2-Pr** showing some LMCT transitions.

We attempted to characterize the electronic structures of compounds **1-Eu** and **2-Sm** using a combination of magnetic studies – electronic paramagnetic resonance (EPR) spectroscopy and superconducting quantum interference device (SQUID) magnetometry – and theoretical analysis. The theoretical approach was based on complete active space self-consistent field spin-orbit-based (CASSCF-SO) calculations, as implemented in OpenMolcas[110,111], where the active spaces are defined by considering seven and five 4f electrons in seven orbitals, respectively (see SI for computational details).

For **1-Eu**, a first calculation considering the ground spin octet and all state-averaged spin sextets, reveals four Kramers doublets (KDs) well-isolated from the rest of the spin-orbit coupled states (Table S8), indicating

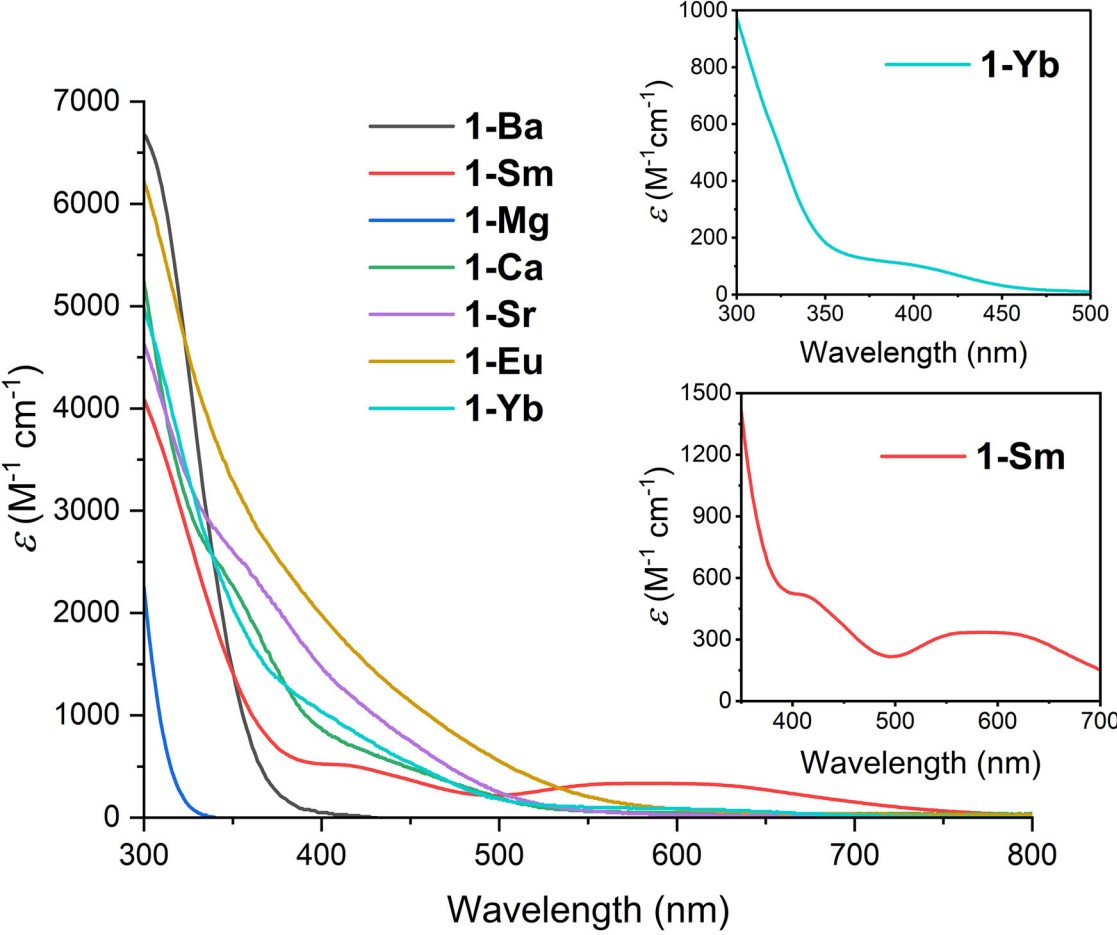

**Fig. 8 | UV-visible spectra for 1-M (0.1 mM, toluene, 300–900 nm).** Insets: UV-visible spectra of **1-Yb** (1.0 mM, toluene, 300–500 nm) and **1-Sm · (THF)₂** (350–700 nm).

that the magnetic response of the compound is dominated by these low-lying states. Inspection of these KDs wavefunctions reveals that they are made up of linear combinations of the ground $^8S_{7/2}$ free ion term and the states coming from the first excited $^6P$ term only, which allows us to simplify the calculation and consider one S = 7/2 and three S = 5/2 states. This in turn permits including dynamic electronic correlation effects by means of the extended multi-state (XMS)[112,113] CASPT2 formalism[114] (see Table S7 for details on the different approaches). Thus, the interaction between the ground $^8S_{7/2}$ state with the excited $^6P$ term states via spin-orbit coupling, leads to magnetic anisotropy and the appearance of four Kramers doublets, which can be parametrised using a zero-field splitting Hamiltonian combined with an electronic Zeeman term (see SI) as

$$\hat{H} = \hat{H}_{Zee} + \hat{H}_{ZFS} = \mu_B \hat{S}\bar{\bar{g}}\vec{B} + \left[ D\left(\hat{S}_z^2 - \frac{\hat{S}^2}{3}\right) + E\left(\hat{S}_x^2 - \hat{S}_y^2\right) \right] \quad (1)$$

where D and E describe the axial and rhombic terms of the anisotropy, with $|D| \geq 3E \geq 0$ (see Table S10). All employed approaches predict susceptibility and magnetisation curves that match well with the experimental values (Fig. 9). We note, however, that the sudden drop in $\chi T$ product could not be explained with this model – the only other option we could think of involves intermolecular antiferromagnetic interactions, which is not supported by the crystal data.

We then proceeded with the characterization of **1-Eu** *via* continuous-wave Electronic Paramagnetic Resonance (cw EPR) spectroscopy. Variable temperature X-band spectra of **1-Eu** recorded between 5 K and 291 K show a similar overall shape, but also display significant variations in the relative intensities across the temperature range (Fig. 10, left), indicating. the

presence of multiple thermally accessible states with different magnetic character. The Q-band spectrum, measured at room temperature, presents a much richer structure, which was used to test the validity of our computational approach. Unfortunately, using the CASSCF-SO obtained parameters, the agreement between experiment and simulation is poor (Table S10, Fig. S86 and S87). However, assuming a model spin Hamiltonian as indicated in Eq. 1, we were able to simulate the Q-band spectrum (Fig. 10, right) while also reproducing the main features of the X-band spectrum at room temperature (Fig. S87, top) using EasySpin[115]. We note that other D and E values could be used to simulate the Q-band spectrum but failed at reproducing the X-band data.

Given the strikingly different parameters set obtained by CASSCF-SO calculations and simulating the EPR spectra, we turned to the molecular geometry looking for an explanation. We note that the presence of extensive crystallographic positional disorder in **1-Eu** and the occurrence of different conformers could be responsible for the difficulty in predicting the correct D and E values. We have tried to address this by performing CASSCF-SO calculations using different geometric models (Table S11), but to no avail. Therefore, at present, we are not able to produce a theoretical model that captures the experimental magnetic data of **1-Eu** in their entirety.

For **2-Sm**, despite all efforts, we were unable to obtain reliable susceptibility or magnetization curves. This is likely due to the small magnetic moment associated with the Sm(III) ion, supported by CASSCF-SO calculations (Table S12) and the electronic structure of the compound will not be discussed further.

## Conclusions

In summary, we have shown that **1-M** (M = Mg–Ba, Eu, Yb) can be readily prepared from deprotonation reaction between dibenzylmetal precursors

**Fig. 9 | Comparison of experimental and calculated susceptibility and magnetisation data for 1-Eu.** Main graph shows susceptibility curves. The inset shows magnetisation curves. Experimental data are displayed using circles. Calculated data are shown using either solid lines (susceptibility curve) or dashed lines (magnetisation curves, XMS-CASPT2 results).

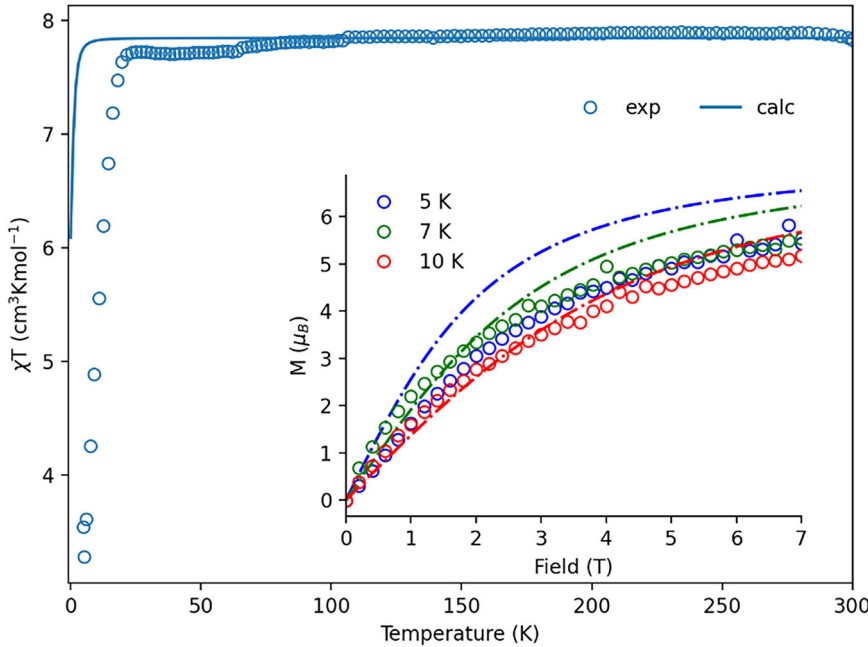

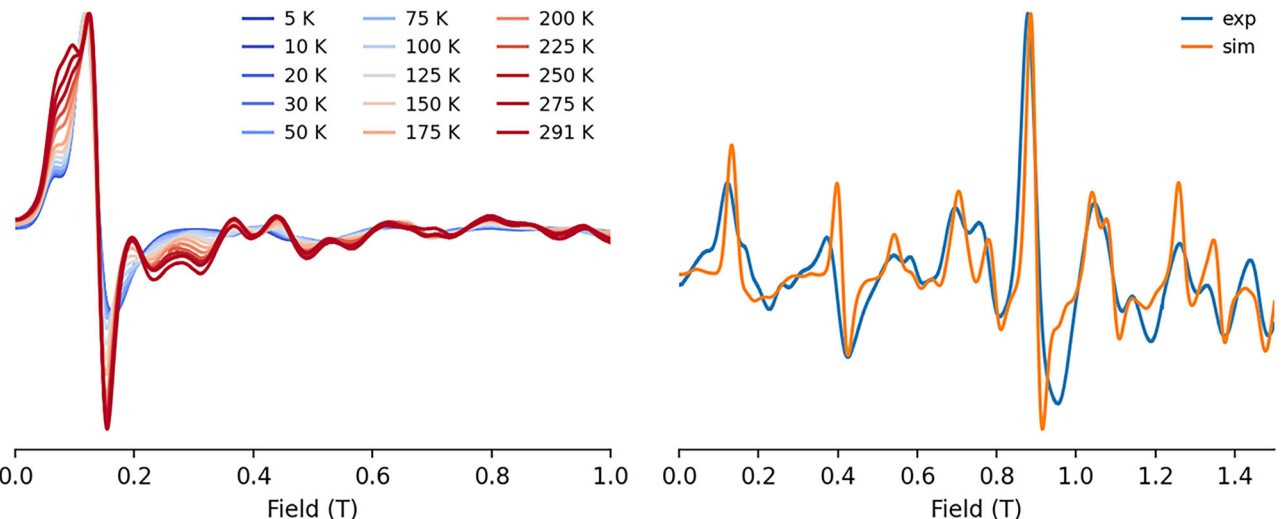

**Fig. 10 | EPR spectra of 1-Eu.** Left: comparison of normalised X-band spectra as a function of temperature. Right: Comparison of normalised experimental and simulated Q-band spectra at 298 K. Simulated as a $S = 7/2$, $g = 1.989$, $D = -7620$ MHz, $E = 300$ MHz.

and the proligand NPC-H$_2$, with the exception of the Sm analogue **1-Sm·(THF)$_2$** that could only be isolated *via* salt elimination reaction between [{K(NPC-H)}$_2$] and SmI$_2$·(THF)$_2$. The latter method was extended successfully to other REI$_3$·(THF)$_x$ salts (RE = Y, La, Pr) to prepare **2-RE** (RE = Y, La, Pr, Sm). This series of compounds displayed familiar trends in their structural and spectroscopic characteristics, particularly with regards to the Group 2 derivatives. The reactivity of **1-Yb**, **2-Y** and **2-La** was tested with HN(SiMe$_3$)$_2$ and HOC$_6$H$_3$$^t$Bu$_2$-2,6. For both sets of reactions, the respective aryloxide complexes – [{Yb(OC$_6$H$_3$$^t$Bu$_2$-2,6)(μ-OC$_6$H$_3$$^t$Bu$_2$-2,6)}$_2$] and [RE(OC$_6$H$_3$$^t$Bu$_2$-2,6)$_3$] – were obtained, whilst reactivity with HN(SiMe$_3$)$_2$ produced the relative amide complex only when using **1-Yb** – i.e., [{Yb[N(SiMe$_3$)$_2$][μ-N(SiMe$_3$)$_2$]}$_2$]. Most notably, reactivity of **1-Yb** was very smooth, leading to complete conversions at mild conditions. This reactivity studies show that such RE methanides could indeed be useful synthetic precursors for the preparation of solvent-free complexes. We have also carried out an extensive spectroscopic characterization of all AE and RE methanides produced in this work. We observed a rough correlation

between ionic radius (an approximation for the ionic character of the bonding) and the $^{13}$C{$^1$H} chemical shift along with the $^1J_{CP}$ coupling constant of the methanide carbon, particularly across the **1-M** series. A similar pattern was observed in the $^{29}$Si{$^1$H} NMR resonance of the adjacent silicon atom. In the case of **1-Yb**, we recorded, to the best of our knowledge, the first $^{171}$Yb NMR of an ytterbium complex with an iminophosphoranomethanide ligand and found that it has a chemical shift ($\delta_{Yb} = 1046.5$ ppm) that would be similar to analogous formamidinate systems, but is less deshielding than β-diketiminates. It is also quite similar to the ytterbium silanide ($\delta_{Yb} = 1044.64$ ppm) reported by Liddle and co-workers[90]. The electronic structure investigation of **1-Eu** reveals notable differences between some of the experimental data and modelling using the ZFS parameters arising from CASSCF-SO calculations. This could be due to significant structural variations that result in complex mixtures of conformers, but despite our best efforts we could not unequivocally solve this problem – this might serve as a word of caution when modelling the magnetic properties of similar compounds. We are extending the use of these ligand systems to other trivalent

REs to further probe differences in bonding regimes across the series, together with investigating the reactivity of these species towards unsaturated substrates.

## Methods
### General experimental methods
THF, $Et_2O$, toluene, hexane, and acetonitrile were passed through columns containing molecular sieves, then stored over molecular sieves (acetonitrile), over a potassium mirror ($Et_2O$, hexane, toluene), or over a sodium mirror (THF) and thoroughly degassed prior to use. For NMR spectroscopy, $C_6D_6$ and $C_7D_8$ were dried by refluxing over potassium, and then vacuum transferred and degassed by three freeze-pump-thaw cycles before use. NMR spectra were recorded on either a Bruker AVIII HD 400 or Bruker AVIII 500 spectrometer operating at 400.07/500.13 ($^1$H), 100.60/125.78 ($^{13}$C{$^1$H}), 79.48 ($^{29}$Si{$^1$H}), 161.98 ($^{31}$P{$^1$H}), or 87.52 ($^{171}$Yb{$^1$H}) MHz. To achieve a greater signal-to-noise ratio, the $^{29}$Si{$^1$H} NMR spectra were acquired with a DEPT24 pulse sequence. NMR spectra were recorded at 298 K unless otherwise stated and were referenced to residual solvent signals in the case of $^1$H and $^{13}$C{$^1$H} experiments, or externally referenced to $H_3PO_4$ ($^{31}$P), $SiMe_4$ ($^{29}$Si), or [Yb($C_5Me_5$)$_2$(THF)$_2$] ($^{171}$Yb). FTIR spectra were recorded on a Bruker Alpha II spectrometer with Platinum-ATR module. Elemental microanalyses were carried out by Elemental Microanalysis Ltd and the Elemental analysis service at London Metropolitan University. $^n$Bu$_2$Mg, $^n$BuLi, trimethylsilyl azide, and diphenylmethylphosphine were used as received. Trimethylsilyl chloride was dried over magnesium turnings and thoroughly degassed before use. $CaI_2$, $SrI_2$, and $BaI_2$ were purchased from Avocado Research Chemicals; these were all baked at 300 °C under vacuum for 4 hours before use. Sm and Yb metal turnings were purchased from Avocado Research Chemicals and used as received. Europium diiodide was purchased from Merk Life Scientific and used as received. Benzylpotassium[116], SmI$_2$(THF)$_2$[117], YbI$_2$(THF)$_2$[117], $N$-trimethylsilylimino-$P,P$-diphenylphosphoranomethane[118] were prepared according to literature procedures. $^{TMS}$NPC-H$_2$ was prepared following previous literature procedures[6]. [Y{N(SiMe$_3$)$_2$}$_3$] and [La{N(SiMe$_3$)$_2$}$_3$] were synthesized following literature procedures[2].

### Synthesis
[Mg($^{TMS}$NPC-H)$_2$] (1-Mg): To a solution of $^{TMS}$NPC-H$_2$ (1.436 g, 4.0 mmol) in hexane (20 mL) was added di-$n$-butylmagnesium (2.2 mL, 2.2 mmol, 1.0 M in heptane) at room temperature with stirring. The solution was allowed to stir overnight, then the resulting white precipitate was filtered off to yield **1-Mg** (1.184 g, 80%, 1.60 mmol) as a white powder. Crystals of **1-Mg** were subsequently obtained upon recrystallization from toluene (ca. 1 g, 4 mL) at room temperature.

$^1$H NMR (400 MHz, 298 K, $C_6D_6$): δ/ppm = 0.08-0.38 (m, 36H, SiC$H_3$), 1.01-1.29 (m, 2H, C$H$-Mg), 7.09-7.18 (m, 12H, $m$-C$_6H_5$ + $p$-C$_6H_5$), 7.80-7.95 (m, 8H, $o$-C$_6H_5$). $^1$H NMR (400 MHz, 378 K, $C_7D_8$): δ/ppm = −0.07 (18H, s, SiC$H_3$), 0.04 (18H, s, SiC$H_3$), 1.05 (2H, d, $^2J_{HP}$ = 15.0 Hz, C$H$-Mg), 7.20 (6H, m, $m$-C$_6H_5$ + $p$-C$_6H_5$), 7.90 (m, 4H, $o$-C$_6H_5$). $^{13}$C{$^1$H} (100 MHz, 298 K, $C_6D_6$): δ/ppm = 2.9 (d, $^3J_{CP}$ = 5.1 Hz, N-Si(CH$_3$)$_3$), 3.5 (d, $^3J_{CP}$ = 3.8 Hz, CH-Si(CH$_3$)$_3$), 17.8 (d, $^1J_{CP}$ = 59.9 Hz, P-CH-Mg), 130.3 (d, $J_{CP}$ = 2.8 Hz, $m$-C$_6H_5$), 130.9 (d, $J_{CP}$ = 2.5 Hz, $m$-C$_6H_5$), 131.4 (d, $J_{CP}$ = 10.6 Hz, $o$-C$_6H_5$), 132.6 (d, $J_{CP}$ = 10.4 Hz, $o$-C$_6H_5$), 136.1 (s, $p$-C$_6H_5$), 137.8 (s, $p$-C$_6H_5$); ipso-C$_6H_5$ not detected. $^{31}$P{$^1$H} (162 MHz, 298 K, $C_6D_6$): δ/ppm = 30.1 ppm; $^{29}$Si{$^1$H} (79 MHz, 378 K, $C_7D_8$): δ/ppm = −8.2 (d, $^2J_{SiP}$ = 5.4 Hz, N-$Si$Me$_3$), −7.8 (s, CH-$Si$Me$_3$). Anal. calcd for $C_{38}H_{58}N_2MgP_2Si_4$: C, 61.55%; H, 7.88%; N, 3.78%. Found: C, 61.02%; H, 7.88%; N, 7.65%. FTIR: $\tilde{\nu}$/cm$^{-1}$ = 469, 490, 529, 587, 613, 694, 722, 747, 826, 919, 1095, 1117, 1244, 1435, 2947. UV/vis (0.2 mM, $C_7H_8$): $\lambda_{max}$/ nm (ε /M$^{-1}$ cm$^{-1}$) = 283 (5980).

### Synthesis of *bis*-methanides [M($^{TMS}$NPC-H)$_2$] (1-M; M = Ca–Ba, Eu, Yb) *via* deprotonation reaction.
A mixture of benzylpotassium (0.543 g, 4.2 mmol) and metal diiodide (2.2 mmol) was dissolved in THF (30 mL) and stirred at room temperature for 5 hours. The THF solvent was then

removed *in vacuo*, and a solution of $^{TMS}$NPC-H$_2$ (1.077 g, 3.0 mmol) in toluene (30 mL) was then added, and the flask swas craped to ensure all dibenzyl metal was in suspension. The reaction mixture was allowed to react overnight, then the precipitate was filtered off and the toluene solvent removed entirely *in vacuo*. The resulting oil was then dried under vacuum ($10^{-2}$ mbar) for at least 6 hours at 50 °C until the pressure stopped dropping, which resulted in a free-flowing powder.

**1-Ca**. Red-orange powder, 1.091 g (1.44 mmol, 96%). Crystals of **1-Ca** were obtained upon recrystallization from toluene (ca. 1 g, 2 mL) at room temperature. $^1$H NMR (400 MHz, 298 K, $C_6D_6$): δ/ppm = 0.06 (18H, s, SiC$H_3$), 0.14 (s, 18H, SiC$H_3$), 0.63 (2H, d, $^2J_{HP}$ = 13.0 Hz, P-C$H$-Ca), 7.09-7.18 (12H, m, $m,p$-C$_6H_5$), 7.82 (8H, dd, $^3J_{HH}$ = 8.0 Hz, $^2J_{HP}$ = 12.0 Hz, $o$-C$_6H_5$). $^{13}$C{$^1$H} (100 MHz, 298 K, $C_6D_6$): : δ (ppm) = 4.0 (d, $^3J_{CP}$ = 4.6 Hz, N-Si(CH$_3$)$_3$), 4.1 (d, $^3J_{CP}$ = 3.8 Hz, CH-Si(CH$_3$)$_3$), 20.7 (d, $^1J_{CP}$ = 78.0 Hz, P-CH-Ca), 128.2 (s, $p$-C$_6H_5$), 130.4 (d, $^3J_{CP}$ = 2.5 Hz, $m$-C$_6H_5$), 132.0 (d, $^2J_{CP}$ = 11.3 Hz, $o$-C$_6H_5$); ipso-C$_6H_5$ not detected. $^{31}$P{$^1$H} (162 MHz, 298 K, $C_6D_6$): δ/ppm = 22.4. $^{29}$Si{$^1$H} (79 MHz, 298 K, $C_6D_6$): δ (ppm) = −11.4 (d, $^2J_{SiP}$ = 1.9 Hz, CH-$Si$Me$_3$), −10.3 (d, $^2J_{SiP}$ = 9.6 Hz, N-$Si$Me$_3$). Anal. calcd for $C_{38}H_{58}N_2CaP_2Si_4\cdot(C_6H_{14})_{0.5}$: C, 61.53%; H, 8.19%; N, 3.50%. Found: C, 61.87%; H, 7.92%; N, 3.00%. FTIR: $\tilde{\nu}$/cm$^{-1}$ = 426, 482, 529, 600, 694, 713, 744, 823, 930, 1123, 1242, 1434, 2945. UV/vis (0.1 mM, $C_7H_8$): $\lambda_{max}$/nm (ε /M$^{-1}$ cm$^{-1}$) = 283 (6900), 341 (2460), 425 (650).

**1-Sr·(THF)$_2$**. Red powder, 0.797 g (0.91 mmol, 78%). Crystals of **1-Sr** were obtained upon recrystallization from toluene (ca. 0.8 g, 1 mL) at −25 °C. $^1$H NMR (400 MHz, 298 K, $C_6D_6$): δ/ppm = 0.12 (18H, s, SiC$H_3$), 0.13 (18H, s, SiC$H_3$), 0.30 (2H, d, $^2J_{HP}$ = 14.3 Hz, P-CH-Sr), 1.26 (br, THF O-C$H_2$), 3.61 (br, THF OCH$_2$C$H_2$), 7.15 (2H, t, $^3J_{HH}$ = 7.1 Hz, $p$-C$_6H_5$), 7.20 (4H, ddd, $^3J_{HH}$ = 6.9 Hz × 2, $^4J_{HP}$ = 1.8 Hz, $m$-C$_6H_5$), 7.89 (4H, dd, $^3J_{HP}$ = 11.7 Hz, $^3J_{HH}$ = 7.1 Hz, $o$-C$_6H_5$). $^{13}$C{$^1$H} (100 MHz, 298 K, $C_6D_6$): δ/ppm = 4.3 (d, $^3J_{CP}$ = 4.0 Hz, CH-SiCH$_3$), 4.4 (d, $^3J_{CP}$ = 4.8 Hz, N-SiCH$_3$), 20.8 (d, $^1J_{CP}$ = 79.5 Hz, Sr-CH), 25.5 (s, THF O-$CH_2$), 68.5 (s, THF O-CH$_2$CH$_2$), 127.9 (s, $p$-C$_6H_5$), 130.0 (d, $^3J_{CP}$ = 2.6 Hz, $m$-C$_6H_5$), 132.2 (d, $^2J_{CP}$ = 10.3 Hz, $o$-C$_6H_5$), 140.7 (d, $^1J_{CP}$ = 84.4 Hz, ipso-C$_6H_5$). $^{31}$P{$^1$H} (162 MHz, 298 K, $C_6D_6$): δ/ppm = 22.3. $^{29}$Si{$^1$H} (79 MHz, 298 K, $C_6D_6$): δ/ppm = −13.0 (d, $^2J_{SiP}$ = 3.5 Hz, CH-$Si$Me$_3$), −10.7 (d, $^2J_{SiP}$ = 10.1 Hz, N-$Si$Me$_3$). Anal. calcd for $C_{38}H_{58}N_2P_2Si_4Sr$: C, 56.71%; H, 7.26%; N, 3.48%. Found: C, 50.00%; H, 6.32%; N, 3.18%; low C and H values were obtained consistently on repeated attempts. FTIR: $\tilde{\nu}$/cm$^{-1}$ = 527, 599, 695, 743, 822, 927, 1031, 1105, 1240, 1434, 1577, 1801, 2946. UV/vis (0.1 mM, $C_7H_8$): $\lambda_{max}$/nm (ε /M$^{-1}$ cm$^{-1}$) = 283 (5200), 355 (2500), 448 (770).

**1-Ba·(THF)$_2$**. Orange powder, 0.584 g (0.63 mmol, 42%). Crystals of **1-Ba** were obtained upon recrystallization from toluene (ca. 0.6 g, 0.5 mL) at −25 °C. $^1$H NMR (400 MHz, 298 K, $C_6D_6$): δ/ppm = 0.08 (2H, d, $^2J_{HP}$ = 11.7 Hz, C$H$-Ba), 0.15 (18H, s, SiC$H_3$), 0.18 (18H, s, SiC$H_3$), 1.29 (4H, m, THF O-CH$_2$C$H_2$), 3.52 (4H, m, THF O-C$H_2$), 7.10-7.21 (12H, m, $m,p$-C$_6H_5$), 7.82 (8H, dd, $^3J_{HP}$ = 11.8 Hz, $^3J_{HH}$ = 7.4 Hz, $o$-C$_6H_5$). $^{13}$C{$^1$H} (100 MHz, 298 K, $C_6D_6$): δ/ppm = 4.5 (d, $^3J_{CP}$ = 3.4 Hz, CH-SiCH$_3$), 4.5 (d, $^3J_{CP}$ = 4.6 Hz, N-SiCH$_3$), 21.6 (d, $^1J_{CP}$ = 88 Hz, Ba-CH), 25.3 (s, THF O-$CH_2$), 68.5 (s, THF O-CH$_2$CH$_2$), 128.6 (s, $p$-C$_6H_5$), 129.6 (d, $^3J_{CP}$ = 2.5 Hz, $m$-C$_6H_5$), 131.7 (d, $^2J_{CP}$ = 10.3 Hz, $o$-C$_6H_5$), 142.3 (d, $^1J_{CP}$ = 84.2 Hz, ipso-C$_6H_5$). $^{31}$P{$^1$H} (162 MHz, 298 K, $C_6D_6$): δ/ppm = 20.3 ppm. $^{29}$Si{$^1$H} (79 MHz, 298 K, $C_6D_6$): δ/ppm = −13.4 (d, $^2J_{SiP}$ = 5.5 Hz, CH-$Si$Me$_3$), −10.3 (d, $^2J_{SiP}$ = 11.5 Hz, N-$Si$Me$_3$). Anal. calcd for $C_{38}H_{58}BaN_2P_2Si_4$: C, 53.41%; H, 6.84%; N, 3.28%. Found: C, 53.18%; H, 7.03%; N, 3.10%. FTIR: $\tilde{\nu}$/cm$^{-1}$ = 478, 527, 596, 695, 743, 820, 929, 1128, 1239, 1433, 2943. UV/vis (1.0 mM, $C_7H_8$): $\lambda_{max}$/nm (ε /M$^{-1}$ cm$^{-1}$) = 418 (133), 300 (6680).

**1-Eu**. Dark red powder, 1.269 g (1.46 mmol, 73%). Crystals of **1-Eu** were obtained upon recrystallization from toluene (ca. 1 g, 2 mL) at −35 °C. $^1$H

NMR (400 MHz, 298 K, $C_6D_6$): δ/ppm = −1.61 (br, $v_{1/2}$ = 5300 Hz), 0.08 (br, $v_{1/2}$ = 120 Hz), 0.51 (br), 0.88 (br), 1.29 (br), 2.11 (br, $v_{1/2}$ = 80 Hz), 2.74 (br), 7.81 (br) ppm. $^{31}P\{^1H\}$ (162 MHz, $C_6D_6$): δ/ppm = 35.3 (br, $v_{1/2}$ = 270 Hz). Evans method ($^1H$ NMR, 298 K, $C_6D_6$): $\mu_{eff}$ = 7.25 $\mu_B$. Anal. calcd for $C_{38}H_{58}N_2EuP_2Si_4$: C, 52.51%; H, 6.73%; N, 3.22%. Found: C, 50.04%; H, 6.60%; N, 2.90%; low C values were obtained consistently on repeated measurements, which we ascribe to incomplete combustion and carbide formation, as previously observed in silicon-rich lanthanide compounds[22,119]. FTIR: $\tilde{v}/cm^{-1}$ = 526, 593, 694, 742, 820, 928, 1102, 1240, 1434, 2943. UV/vis (0.1 mM, $C_7H_8$): $\lambda_{max}$/nm (ε /$M^{-1}$ $cm^{-1}$) = 288 (6700).

**1-Yb**. Dark red powder, 1.353 g (1.52 mmol, 76%). Crystals of **1-Yb** were obtained upon recrystallization from toluene (ca. 1 g, 2 mL) at –25 °C. $^1H$ NMR (400 MHz, 298 K, $C_6D_6$): δ/ppm = 0.07 (18H, s, SiC$H_3$), 0.15 (18H, s, SiC$H_3$), 0.77 (2H, d, $^2J_{HP}$ = 12.7 Hz, Yb-C$H$), 7.00-7.08 (8H, m), 7.10-7.21 (4H, m), 7.82 (8H, dd, $^3J_{HP}$ = 11.9 Hz, $^3J_{HH}$ = 7.1 Hz, o-$C_6H_5$). $^{13}C\{^1H\}$ (100 MHz, 298 K, $C_6D_6$): δ/ppm = 3.97 (d, $^3J_{CP}$ = 4.6 Hz, N-SiC$H_3$), 4.02 (d, $^3J_{CP}$ = 3.7 Hz, CH-SiC$H_3$), 24.2 (d, $^1J_{CP}$ = 78.7 Hz, Yb-$CH$), 130.3 (d, $^4J_{CP}$ = 2.5 Hz, p-$C_6H_5$), 130.4 (d, $^3J_{CP}$ = 3.0 Hz, m-$C_6H_5$), 131.1 (d, $^2J_{CP}$ = 10.1 Hz, o-$C_6H_5$), 131.9 (d, $^1J_{CP}$ = 10.4 Hz, ipso-$C_6H_5$). $^{31}P\{^1H\}$ (162 MHz, 298 K, $C_6D_6$): δ/ppm = 19.7. $^{29}Si\{^1H\}$ (79 MHz, 298 K, $C_6D_6$): δ/ppm = −12.5 (d, $^2J_{SiP}$ = 2.6 Hz, CH-$Si$Me₃), −10.4 (d, $^2J_{SiP}$ = 9.8 Hz, N-$Si$Me₃). $^{171}Yb\{^1H\}$ (88 MHz, 298 K, $C_6D_6$): δ/ppm = 1046.5 (v½ = 290 Hz). Anal. calcd for $C_{38}H_{58}N_2P_2Si_4Yb·(C_6H_{14})_{0.5}$: C, 52.76%; H, 7.02%; N, 3.00%. Found: C, 52.36%; H, 6.80%; N, 2.88%. FTIR: $\tilde{v}/cm^{-1}$ = 527, 597, 694, 712, 743, 824, 926, 1103, 1242, 1434, 2947. UV/vis (0.1 mM, $C_7H_8$): $\lambda_{max}$/nm (ε /$M^{-1}$ $cm^{-1}$) = 295 (5200), 404 (1040), 600 (110).

**2-Sm**. A mixture of benzylpotassium (0.543 g, 4.2 mmol) and $SmI_2(THF)_2$ (2.2 mmol) were dissolved in THF (30 mL) and stirred at room temperature for 5 hours. The THF solvent was then removed in vacuo, and a solution of $^{TMS}NPC-H_2$ (1.077 g, 3.0 mmol) in toluene (30 mL) was then added, and the flask scraped to ensure all dibenzyl metal was in suspension. The reaction mixture was allowed to react overnight, then the precipitate was filtered off and the toluene solvent removed entirely in vacuo. The resulting oil was then dried under vacuum for at least 6 hours at 50 °C until the pressure stopped dropping, which resulted in a free-flowing dark powder. This was recrystallized from toluene at –25 °C, yielding a small crop (ca. 15 mg) of crystalline **2** sufficient only for X-ray studies and spectroscopic analysis (vide infra for full characterization).

**Synthesis of [Sm($^{TMS}$NPC-H)₂(THF)₂] (1-Sm·(THF)₂).** [{K($^{TMS}$NPC-H)}₂] (1.500 g, 3.8 mmol) and $SmI_2(THF)_2$ (1.034 g, 1.9 mmol) were added to a flame-dried Schlenk fitted with a stir bar. THF (20 mL) was added and the mixture was stirred at room temperature for 16 hours. Volatiles were removed in vacuo and the solid residue was extracted with hexane (2 × 20 mL). The solution was stored at –25 °C, affording **1-Sm·(THF)₂** as dark blue crystals (1.051 g, 1.0 mmol, 53%). $^1H$ NMR (400 MHz, 298 K, $C_6D_6$): δ/ppm = −2.47, −2.17 (broad, $v_{1/2}$ = 38 Hz, –0.45 (broad, $v_{1/2}$ = 21 Hz), 0.02 (s), 0.11 (broad, $v_{1/2}$ = 70 Hz), 0.33, 0.89, 1.07 (broad, $v_{1/2}$ = 31 Hz), 1.23, 2.03 (broad, $v_{1/2}$ = 21 Hz), 4.70, 5.23 (broad, $v_{1/2}$ = 74 Hz), 5.68 (broad, $v_{1/2}$ = 23 Hz), 6.01 (broad, $v_{1/2}$ = 24 Hz), 7.00 (broad, $v_{1/2}$ = 41 Hz) 8.22 (broad, $v_{1/2}$ = 21 Hz), 14.17 (broad, $v_{1/2}$ = 132 Hz). Evans method ($^1H$ NMR, 298 K, $C_6D_6$): $\mu_{eff}$ = 3.58 $\mu_B$. Anal. calcd for $C_{46}H_{74}N_2OP_2Si_4Sm$: C, 54.61%; H, 7.37%; N, 2.77%. Found: C, 51.35%; H, 7.48%; N, 2.24%; low C values were obtained consistently on repeated measurements, which we ascribe to incomplete combustion and carbide formation, as previously observed in silicon-rich lanthanide compounds[22,119]. FTIR: $\tilde{v}/cm^{-1}$ = 2956 (w), 2875 (w), 1473 (w), 1434 (w), 1238 (m), 1123 (s), 1107 (s), 1035 (m), 998 (w), 923 (w), 880 (w), 845 (s), 822 (s), 768 (w), 744 (s), 697 (s), 658 (m), 599 (m), 527 (s), 484 (m). UV/vis (0.1 mM, $C_4H_8O$): $\lambda_{max}$/nm (ε /$M^{-1}$ $cm^{-1}$) = 249 (9469), 410 (517), 588 (335).

**Synthesis of *tris*-methanides [RE($^{TMS}$NPC-H)₃] (2-RE; RE = Y, La, Pr, Sm) *via* salt elimination.** [{K($^{TMS}$NPC-H)}₂] (1.191 g, 3.0 mmol) and $REI_3(THF)_x$ (1.0 mmol, RE = La, Pr, x = 4; Ln = Y, Pr, Sm, x = 3.5) were added to a flame-dried Schlenk flask fitted with a stir bar. THF (30 mL) was added, and the reaction mixture was stirred overnight. The solvent was then removed in vacuo, and the residue extracted with toluene. The solution was concentrated to ca. 3 m,L whereupon crystals formed at room temperature. A second crop was reliably obtained by cooling to –25°C.

**2-Y**. Colourless crystals (0.372 g, 32%, 0.32 mmol). $^1H$ NMR (500 MHz, 378 K, $C_7D_8$): δ/ppm = −0.13 (16H, s, SiC$H_3$), 0.03 (21H, s, SiC$H_3$), 0.21 (18H, s, SiC$H_3$), 1.57 (3H, d, $^2J_{HP}$ = 14.6 Hz, Y–C$H$), 7.09-7.16 (12H, m, $C_6H_5$), 7.56-7.67 (6H, m, $C_6H_5$), 7.74-7.91 (12H, m, $C_6H_5$). $^{13}C\{^1H\}$ (100 MHz, 298 K, $C_6D_6$): δ/ppm = 4.63 (d, $^3J_{CP}$ = 5.1 Hz, N–SiC$H_3$), 6.29 (d, $^3J_{CP}$ = 3.9 Hz, CH–SiC$H_3$), 21.6 (dd, $^1J_{CP}$ = 47 Hz, $^1J_{CY}$ = 8.6 Hz Y–C$H$), 127.7 (s, p-$C_6H_5$), 129.8 (d, $^3J_{CP}$ = 2.8 Hz, m-$C_6H_5$), 130.6 (d, $^2J_{CP}$ = 10.1 Hz, o-$C_6H_5$), 139.0 (d, $^1J_{CP}$ = 95.2 Hz, ipso-$C_6H_5$). $^{31}P\{^1H\}$ (162 MHz, 298 K, $C_6D_6$): δ/ppm = 29.9 (d, $^2J_{PY}$ = 8.2 Hz), 31.7 (d, $^2J_{PY}$ = 8.2 Hz), 32.2 (d, $^2J_{PY}$ = 8.2 Hz), 33.6 (d, $^2J_{PY}$ = 8.2 Hz). $^{29}Si\{^1H\}$ (79 MHz, 378 K, $C_7D_8$): δ/ppm = −7.23 (d, $^2J_{SiP}$ = 4.0 Hz, CH-$Si$Me₃), −4.90 (d, $^2J_{SiP}$ = 2.9 Hz, N-$Si$Me₃). Anal. calcd for $C_{57}H_{87}N_3P_3Si_6Y$: C, 58.78%; H, 7.53%; N, 3.61%. Found: C, 58.96%; H, 7.41%; N, 3.19%. FTIR: $\tilde{v}/cm^{-1}$ = 411, 463, 498, 528, 590, 613, 651, 695, 728, 744, 824, 902, 1002, 1019, 1036, 1101, 1244, 1435, 1482, 2895, 2955. UV/vis (0.1 mM, $C_7H_8$): $\lambda_{max}$/nm (ε /$M^{-1}$ $cm^{-1}$) = 290 (50000).

**2-La**. Colourless crystals (0.455 g, 37%, 0.37 mmol). $^1H$ NMR (400 MHz, 298 K, $C_6D_6$): δ/ppm = 0.14 (33H, s, SiC$H_3$), 0.32 (21H, s, SiC$H_3$), 1.14 (3H, d, $^2J_{HP}$ = 15.8 Hz), 7.04-7.10 (12H, m, $C_6H_5$), 7.86 (6H, t, $^3J_{HH}$ = 8.5 Hz, p-$C_6H_5$), 7.93-8.05 (12H, m, $C_6H_5$). $^{13}C\{^1H\}$ (100 MHz, 298 K, $C_6D_6$): δ/ppm = 4.66 (br, $v_{1/2}$ = 12 Hz), 5.20 (4.0 Hz), 130.7 (d, $J_{CP}$ = 10.6 Hz), 133.9 (d, $J_{CP}$ = 10.6 Hz). $^{31}P\{^1H\}$ (162 MHz, 298 K, $C_6D_6$): δ/ppm = 27.8. $^{29}Si\{^1H\}$ (79 MHz, 298 K, $C_6D_6$): no signals observed. FTIR: $\tilde{v}/cm^{-1}$ = 412, 462, 489, 527, 583, 606, 642, 693, 709, 742, 757, 822, 906, 1000, 1027, 1057, 1101, 1243, 1435, 1481, 2892, 2950. Satisfactory elemental analyses could not be obtained due to the persistent presence of impurities (proligand), despite repeated recrystallizations.

**2-Pr**. Light green crystals (0.793 g, 65%, 0.65 mmol). $^1H$ NMR (400 MHz, $C_6D_6$, 298 K) δ/ppm = −14.00 (br, $v_{1/2}$ = 72 Hz), −13.07 (br, $v_{1/2}$ = 84 Hz), −8.46 (br, $v_{1/2}$ = 73 Hz), −7.36 (br, $v_{1/2}$ = 114 Hz), −5.85 (br, $v_{1/2}$ = 73 Hz), −4.51 (br, $v_{1/2}$ = 24 Hz), 0.01 (s), 0.07 (s), 0.15 (s), 0.34 (s), 2.11 (s), 2.20 (br, $v_{1/2}$ = 79 Hz), 2.69 (br, $v_{1/2}$ = 106 Hz), 7.06 (br, $v_{1/2}$ = 113 Hz), 7.59 (br, $v_{1/2}$ = 26 Hz), 7.89 (br, $v_{1/2}$ = 25 Hz), 8.57 (br, $v_{1/2}$ = 155 Hz), 9.35 (br, $v_{1/2}$ = 17 Hz), 9.75 (br, $v_{1/2}$ = 17 Hz), 10.01 (br, $v_{1/2}$ = 18 Hz), 10.60 (br, $v_{1/2}$ = 17 Hz), 12. 20 (br, $v_{1/2}$ = 28 Hz), 17.26 (br, $v_{1/2}$ = 49 Hz), 22.25 (br, $v_{1/2}$ = 83 Hz), 24.83 (br, $v_{1/2}$ = 76 Hz). $^{31}P\{^1H\}$ (162 MHz, 298 K, $C_6D_6$): δ/ppm = −187.4 ($v_{1/2}$ = 113 Hz), −121.1 ($v_{1/2}$ = 65 Hz), −88.7 ($v_{1/2}$ = 115 Hz), −63.8 ($v_{1/2}$ = 88 Hz). Evans method ($^1H$ NMR, 298 K, $C_6D_6$): $\mu_{eff}$ = 3.19 $\mu_B$. Satisfactory elemental analyses could not be obtained despite repeated attempts on freshly recrystallized samples.

**2-Sm**. Yellow crystals (0.408 g, 33%, 0.330 mmol). $^1H$ NMR (400 MHz, 298 K, $C_7D_8$): δ/ppm = −12.60 (3H, br, $v_{1/2}$ = 242 Hz), −1.70 (27H, s), −1.04 (27H, s), 7.61-7.72 (9H, m, $C_6H_5$), 7.75 (3H, t, $J_{HH}$ = 7.5 Hz, $C_6H_5$), 7.90 (5H, t, $J_{HH}$ = 7.5 Hz, $C_6H_5$), 9.09 (5H, d, $J_{HH}$ = 7.1 Hz, $C_6H_5$), 9.68 (3H, br, m, $C_6H_5$), 10.23 (5H, d, $J_{HH}$ = 6.2 Hz, $C_6H_5$). $^{13}C\{^1H\}$ (100 MHz, 298 K, $C_6D_6$): δ/ppm = 1.87 (s, SiC$H_3$), 3.80 (s, SiC$H_3$), 125.7, 128.7, 128.8, 129.0, 129.3, 131.0, 131.1, 131.4, 132.2, 135.2, 137.2. $^{31}P\{^1H\}$ (162 MHz, 298 K, $C_6D_6$): δ/ppm = 68.0 (1 P, br, $v_{1/2}$ = 169 Hz), 73.3 (2 P, br, $v_{1/2}$ = 186 Hz). $^{29}Si\{^1H\}$ (79 MHz, 298 K, $C_6D_6$): δ/ppm = −13.3 (d,

$^2J_{SiP}$ = 21.8 Hz, N-$Si$Me$_3$), 0.29 (d, $^2J_{SiP}$ = 5.9 Hz, CH-$Si$Me$_3$). Evans method ($^1$H NMR, 298 K, C$_6$D$_6$): $\mu_{eff}$ = 2.98 $\mu_B$. Satisfactory elemental analyses could not be obtained despite repeated attempts on freshly recrystallized samples. FTIR: $\tilde{\nu}$/cm$^{-1}$ = 409, 463, 496, 528, 588, 611, 649, 693, 728, 744, 823, 901, 1002, 1024, 1038, 1099, 1244, 1435, 1481, 2895, 2955. UV/vis (0.1 mM, C$_7$H$_8$): $\lambda_{max}$/nm ($\varepsilon$ /M$^{-1}$ cm$^{-1}$) = 290 (5000), 718 (39), 876 (72).

**Reactivity between 1-Yb and HN(SiMe$_3$)$_2$. 1-Yb** (53.8 mg, 0.060 mmol) was added to a J Young's NMR tube and dissolved in C$_6$D$_6$ ( ~ 0.6 mL). HN(SiMe$_3$)$_2$ (19.5 mg, 25.3 μL, 0.12 mmol) was added *via* syringe. NMR spectra were recorded after 24 hours, showing quantitative formation of NPC-H$_2$ and complete conversion to [{Yb[N(SiMe$_3$)$_2$][μ-N(SiMe$_3$)$_2$]}$_2$].

**Reactivity between 1-Yb and 2,6-$^t$Bu$_2$C$_6$H$_3$OH. 1-Yb** (55.0 mg, 0.062 mmol) and 2,6-$^t$Bu$_2$C$_6$H$_3$OH (25.5 mg, 0.12 mmol) were added to a J Young's NMR tube and dissolved in C$_6$D$_6$ ( ~ 0.6 mL). NMR spectra were recorded after 48 hours, showing quantitative formation of NPC-H$_2$ and conversion to [{Yb(OC$_6$H$_3$$^t$Bu$_2$-2,6)(μ-OC$_6$H$_3$$^t$Bu$_2$-2,6)}$_2$].

**Reactivity between 2-Y and HN(SiMe$_3$)$_2$. 2-Y** (31.6 mg, 0.027 mmol) was added to a J Young's NMR tube and dissolved in C$_6$D$_6$ (~0.6 mL). HN(SiMe$_3$)$_2$ (13.14 mg, 17.1 μL, 0.081 mmol) was added *via* syringe. NMR spectra were recorded after 16 hours at room temperature, showing almost no conversion. The reaction was heated at 80 °C for 4 hours, showing formation of NPC-H$_2$ and no conversion to [Y{N(SiMe$_3$)$_2$}$_3$].

**Reactivity between 2-Y and 2,6-$^t$Bu$_2$C$_6$H$_3$OH. 2-Y** (36.0 mg, 0.031 mmol) and 2,6-$^t$Bu$_2$C$_6$H$_3$OH (19.0 g, 0.093 mmol) were added to a J Young's NMR tube and dissolved in C$_6$D$_6$ ( ~ 0.6 mL). NMR spectra were recorded after 16 hours at room temperature, showing almost no conversion. The reaction was heated at 80 °C for 7 hours, showing formation of NPC-H$_2$ and no conversion to [Y(OC$_6$H$_3$$^t$Bu$_2$-2,6)$_3$].

**Reactivity between 2-La and HN(SiMe$_3$)$_2$. 2-La** (43.2 mg, 0.036 mmol) was added to a J Young's NMR tube and dissolved in C$_6$D$_6$ (~0.6 mL). HN(SiMe$_3$)$_2$ (17.22 mg, 22.4 μL, 0.107 mmol) was added *via* syringe. NMR spectra were recorded after 16 hours at room temperature, showing no conversion. The rection was heated at 80 °C for 24 hours (NMR spectra recorded at 4, 7, 12 and 24 h), showing formation of NPC-H$_2$ and no conversion to [La{N(SiMe$_3$)$_2$}$_3$], together with one major unidentified species ($\delta_P$ 24.5 ppm).

**Reactivity between 2-La and 2,6-$^t$Bu$_2$C$_6$H$_3$OH. 2-La** (41.5 mg, 0.034 mmol) and 2,6-$^t$Bu$_2$C$_6$H$_3$OH (21.1 mg, 0.102 mmol) were added to a J Young's NMR tube and dissolved in C$_6$D$_6$ (~0.6 mL). NMR spectra were recorded after 16 hours at room temperature, showing almost no conversion. The reaction was heated at 80 °C for 24 hours (NMR spectra recorded at 5, 8, 13, and 24 h), showing formation of NPC-H$_2$ and no conversion to [La(OC$_6$H$_3$$^t$Bu$_2$-2,6)$_3$], together with one minor unidentified species ($\delta_P$ 24.5 ppm).

## Data availability

Additional research data supporting this publication are available from Figshare at https://doi.org/10.25392/leicester.data.28001921. All crystallographic data have been deposited with the Cambridge Crystallographic Data Centre (CCDC 2408361-2408372, 2408465 and 2431369). This information can be obtained free of charge from www.ccdc.cam.ac.uk/data_request/cif.

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

## Acknowledgements
We thank the University of Leicester and the College of Life Sciences and Engineering for a PhD scholarship (M.P.S.) and the Engineering and Physical Sciences Research Council (EP/W00691X/1) for postdoctoral funding (Y.L.) and research funding (F.O.). X-ray diffraction and NMR at Leicester were supported by the Engineering and Physical Sciences Research Council (EP/V034766/1 and EP/W02151X/1). The authors would like to acknowledge the NMR Facility in the School of Chemistry at the University of Leicester for NMR experiments. D. R. thanks the Basque Government for the IT1584-22 grant. The authors thank for technical and human support provided by SGIker (UPV/EHU/ ERDF, EU).

## Author contributions
F.O. provided the original concept. M.P.S. and Y.L. synthesized and characterized the compounds. M.P.S., Y.L., E.A. and S.S.A.X.D. carried out supporting synthetic/characterization work. M.P.S., Y.L. and F.O. carried out the single-crystal X-ray diffraction analysis. R.R.H. and A.K. carried out NMR spectroscopic analysis. D.R. and L.L. collected and interpreted magnetic data. D.R. performed the calculations. F.O., D.R. and M.P.S. wrote the manuscript with contributions from all authors. F.O. and D.R. supervised the project. M.P.S. and Y.L. contributed equally.

## Competing interests
The authors declare no competing interests.
