## [Transparent Peer Review file · Communications Chemistry]

Synthesis, characterization and reactivity of a series of alkaline earth and rare earth iminophosphoranomethanide complexes

Corresponding Author: Dr Fabrizio Ortu

Version 0:

Reviewer comments:

Reviewer #1

(Remarks to the Author)

In this manuscript, Stevens et al. show the synthesis of a series of alkaline earth (Ca, Sr, Ba) and rare earth (Y, La, Pr, Eu, Yb) complexes with an iminophosphoranomethanide ligand. All complexes are structurally characterised by single crystal diffraction, NMR and UV/Vis/NIR spectroscopies and for one of the obtained Eu complexes, EPR studies are reported. These are complemented by a CASSCF study revealing the complexity of interpreting the experimental EPR spectra for this complex, which also serves as a note of caution for other systems which could display different problems. With the present work, the authors expand their previous study on alkali compounds of the iminophosphoranomethanide ligand to alkali earth and rare earth metals, demonstrating the broader applicability of this ligand framework. Additionally, it is shown that three of the rare earth complexes could serve as protonolysis precursors for the synthesis of solvent-free rare earth precursors in an NMR study. As these types of precursors are still limited, the current work of Stevens et al. will surely have an impact on further studies in this field by others.

The overall presentation and scope of the manuscript is good, and I feel it would be suitable for publication in Communications Chemistry after some improvement as per the following more detailed comments.

Introduction:

- Line 32: there is a word (e.g. "compounds") missing after "The chemistry of AE and RE alkyl"
- Figure 1: It would be helpful to the reader if underneath each general structure the metals were listed, for which this complex was reported.
- I am missing some explanation on how the metals were chosen in your study, especially for the rare earths?

Results and Discussion:

- Scheme 1: $M_2(\text{THF})_n$ on the arrows in the left half should be x instead of n
- Reactivity studies: Please add some discussion on why the study was done only with 1-Yb, 2-Y and 2-La, what about 1-Eu, 2-Pr and 2-Sm?
- The NMR spectra of the reactivity studies in the SI are too small to tell if the claimed products were actually formed, please add some information on the identification of the reaction products in the main manuscript and annotate the spectra in figures S59, S61, S64 and S67. I also didn't find the ^1H NMR spectrum for the reaction of 1-Yb and $\text{HN}(\text{SiMe}_3)_2$ as well as the phenol, please add these.
- NMR discussion: p. 7, l. 169: probably a typo in " $^{31}\text{P}\{^1\text{H}\}$ " in the discussion of absence of P-H coupling. Did you measure a ^{31}P NMR spectrum? If so, then please add it to the SI, as currently only the $^{31}\text{P}\{^1\text{H}\}$ spectra are reported.
- P. 9, l. 209: please add the theoretical value for μ
- P. 10, l. 246: some Word error, please check
- P. 13, l. 304: please be more specific what you mean by "more open nature of the NPC-H ligand...".
- P. 14, l. 315: should be Figure 5, not 6
- P. 16, l. 346: should be Figure 6, not 7
- Figure 6: the wavelength unit should read nm, not cm^{-1} . Can you extend the displayed range in the inset to show the discussed f-d transition at 600 nm?
- P. 17, l. 365: Please comment on why the study on the electronic structure was restricted to 1-Eu and 2-Sm. I also suggest adding a sentence to the beginning of this paragraph stating that you performed SQUID and EPR experiments as well.
- Please add the discussion on the results of the electronic structure calculations for 2-Sm to the main manuscript.
- Figure 7: For which compound is this data?
- Are the insights for 1-Eu regarding the mixture of conformers in the EPR relevant for the SQUID measurement?

Experimental section:

- P. 22, l. 502: please specify how high the vacuum was for drying.
- Crystallisation conditions for all complexes: at which temperature where the recrystallizations performed?

- P. 25, l. 574: which colour was the obtained powder?

Supporting information:

- P. 1: please show the ^1H , $^{31}\text{P}\{^1\text{H}\}$ and $^{13}\text{C}\{^1\text{H}\}$ spectra of NPC-H2 as well.

- For the diamagnetic complexes please add integrals to the ^1H NMR spectra

- Fig. S24: Was this spectrum measured on 1-Ca or 1-Ca-THF? The structure shows a THF, but the caption says 1-Ca.

- Fig. S25: The structure shows only one THF, but the compound is labelled as 1-Sr(THF)₂, please double check. Is the number of coordinated THF which can be deduced from the ^1H NMR spectrum the same as in the crystal structure?

- Fig. S47: Please add the peak assignment and the structure.

- Fig. S49: Please add the peak assignment and the structure.

- P. 27, top spectrum: The caption is missing.

- Fig. S58: what are the additional signals observed at ca. 25 ppm starting at 4 hours? Same in Figure S60 and S62

- Fig. S59, S61, S64 and S67: as mentioned above, please assign the signals of the products

Reviewer #2

(Remarks to the Author)

In this work "Synthesis, characterization and reactivity of a series of alkaline earth and rare earth iminophosphoranomethanide complexes," the authors report a series of complexes using a methanide ligand $\{\text{CH}(\text{SiMe}_3)\text{P}(\text{Ph})_2=\text{NSiMe}_3\}^-$ (NPC-H) in the preparation and characterization of alkaline earth and rare earth complexes. The Alkaline earth complexes reported are bismethanide complexes $[\text{M}(\text{NPC-H})_2(\text{THF})_x]$ (1-M·(THF)_x; M = Mg, x = 0; M = Ca, x = 0, 1; M = Sr, x = 0, 2; M = Ba, x = 2).

The rare earth complexes described are bismethanide complexes $[\text{M}(\text{NPC-H})_2(\text{THF})_x]$ (1-M·(THF)_x; M = Eu, Yb, x = 0), the oxidation product $[\text{Sm}(\text{NPC-H})_3]$, and the tris-methanides, $[\text{RE}(\text{NPC-H})_3]$ (2-RE; RE = Y, La, Pr). The electronic structure and calculations are done for $[\text{Eu}(\text{NPC-H})_3]$? but this complex is not completely characterized.

No crystal structure of $[\text{Y}\{\text{C}_6\text{H}_4\text{P}(\text{Ph})_2=\text{NnBu}\}_3]$ was obtained and so this structure was not completely characterized either.

Thus the focus really seems to be the alkaline earth complexes and Sr which are in more detail?

This work appears to overlap with previous work by some of the same authors, as reported in Polyhedron 2024 (reference 6.)

Many complexes of beta-diketamates are reported and would make for an interesting comparison.

This work appears to lack focus or to be incomplete. What trend or utility are the authors trying to demonstrate?

While this does appear to be a great deal of work, and I commend them on their detailed characterization, the authors have not done as great a job supporting their interest in this area nor do they give a compelling argument to its novelty. The paper is a detailed experimental report and does not have an introduction describing other interest in this area or why these compounds are notable.

This work should include more introduction about other research in the area. There are only 2 references from 2024 in what is arguably a wide field of lanthanide complexes. There is only one reference - by the corresponding author referring to a book chapter reviewing interest in the area.

There is not good comparisons as to how these structures compare to related complexes of the lanthanides.

The synthesis is done by protonolysis or salt elimination so not particularly noteworthy. The focus should be on emissions or materials characterization then.

Its not clear to me what the authors are trying to show with the ^1H VT-NMR study of 1-Mg at 25 – 105°C.

The Yb NMR data is interesting and unique - but even if no other Yb data for closely related complexes exists - how does this compare to other complexes?

The Eu complex seems an outlier as it is not isolated and structurally characterized. This should have been left out or in a separate theoretical chemistry article.

Reviewer #3

(Remarks to the Author)

The manuscript describes the synthesis and characterisation of a series of alkaline earth and lanthanide complexes of a silicon- and phosphorus-stabilised carbanion. In general, the work is well done and well described. Some of the compounds are not isolated cleanly, which is a shame, but sufficient evidence of composition is provided. The characterisation methods are very comprehensive, but the data do not always correspond (see below). The authors may wish to consider editing the MS to remove inconsistent calculation data, focusing on the extensive experimental evidence that they have. I recommend acceptance of this article, after the foregoing and the following points have been addressed:

Abstract and page 2 line 48: Protonolysis refers to the transfer of a proton from A to B. As written, the correct term for the described reaction is deprotonation.

Page 3 line 67: "In addition...AlCl₂." This is not a complete sentence – please correct.

Page 3 line 62 and further: Are the transition and p-block complexes relevant here? Better examples would be the complexes of (Me₃Si)₂C{P(BH₃)Me₂]}- with Yb(II) and Sm(II) (Dalton Trans. 2010, 39, 6705), or Mg-Ba (Inorg. Chem. 2007, 46, 4320), or {Ph₂P(BH₃)₂CH₂}₂Ca(THF)(Dalton Trans 2009, 2951) – these would appear to be more comparable to the reported compounds.

Page 4, line 91: Is it not more likely that the Sm(III) product was produced by exposure of the Sm(II) species to oxygen? Was there any evidence for the formation of Sm metal (dark ppt)? Why was salt metathesis not used to access Sm(II) compounds?

Page 7: Discussion of the VT NMR spectra of 1-Mg would be aided by including a suitable figure in the text (rather than the SI). The conclusions drawn from these spectra are rather vague – can something more be deduced?

Page 8: Why is the 171Yb NMR signal broad? Was a VT experiment attempted?

Page 16, line 351: "This is due to the greater 352 ionicity of these bonds, thereby lowering the energy of the transition into the visible region." This statement requires more explanation and a reference.

Page 20: The discussion of the EPR and calculation data is rather poor. The EPR data are experimental and so should be treated as evidence. If the calculations do not correspond with these EPR data, then it must be assumed that the calculations are in error. The explanation of the data as arising from multiple conformers seems unrealistic and should be removed.

Reviewer #4

(Remarks to the Author)

The crystallography has been done well with details of the refinements included in most of the CIFs. The answers to the ALERT A and B's are all reasonable. The crystal structure all support the finding of the authors.

Correction:

Page 10 2.2 Xray crystallography, first line:
....all Error! Reference source not found.

Page 19

Figure 10 is mentioned a couple of times but there is no figure 10. I think it should be figure 9 as there is a figure 9re:.

SI says:"Crystals were examined using a Bruker D8 Quest diffractometer with a Photon III detector"

However, CIF info does not match this in: 1-Ca-THF, 1-Mg-C2/c, 1-Sr, 1-Sr-THF, 2-Yb please correct either the SI or the CIF's. 2-Y appears to have been collect on a Rigaku instrument again correct the SI or CIF.

2-Yb the Centre of Gravity not Within Unit Cell please move the molecule into the unit cell.

Comments and question:

Why does CHECKCIF report large numbers of reflections missing such as 1_Ca where 319 reflections (1-Ca-THF) 146, 1-Sr 201, 2-Sm 128 and 2-Y 146) are reported missing. Is this a result of the data collection strategy or processing issues. This is much high than I have seen before.

Were global RIGU comments really needed to get a good refinement in several of the refinements?

1-Eu why did you not just use displacement restraints to fix the ALERT B, you seem happy to use them in the other refinements.

1-Ba it would have been more consistent to ride the hydrogen displacement parameter like it the other refinements rather than leaving them to freely refine.

Version 1:

Reviewer comments:

Reviewer #1

(Remarks to the Author)

The authors have a done a good job at addressing my remarks and questions and I have only one small remaining

comment: In Scheme 1 in the description underneath 1-M there's still one "n =..." which should read "x = ...". Other than that I'm happy to recommend the manuscript to be accepted for publication.

Reviewer #2

(Remarks to the Author)

The authors have put in considerable effort to clarify their project details. This is most notable in the improved explanations and the additional references incorporated in the introduction and the description.

The corrections to Figures 1, 6, and 7 have helped improve the text. The addition of the ^{31}P NMR to the SI was also informative.

I don't think it's necessary to include the symmetry operations in the figure caption for Figure 7.

The comparisons to references 116 and 117 are helpful.

Overall I think the authors have done a great deal of additional work and have satisfactorily addressed the reviewers comments.

Reviewer #3

(Remarks to the Author)

I am happy that the extensive suggestions made by the reviewers have been adequately addressed and that this manuscript may now be accepted for publication.

Reviewer #4

(Remarks to the Author)

I am happy with the corrections and explanations to my comments. The Crystallography is to a high enough standard to support the conclusion in the manuscript.

Reviewer #1 (Remarks to the Author):

In this manuscript, Stevens et al. show the synthesis of a series of alkaline earth (Ca, Sr, Ba) and rare earth (Y, La, Pr, Eu, Yb) complexes with an iminophosphoranomethanide ligand. All complexes are structurally characterised by single crystal diffraction, NMR and UV/Vis/NIR spectroscopies and for one of the obtained Eu complexes, EPR studies are reported. These are complemented by a CASSCF study revealing the complexity of interpreting the experimental EPR spectra for this complex, which also serves as a note of caution for other systems which could display different problems. With the present work, the authors expand their previous study on alkali compounds of the iminophosphoranomethanide ligand to alkali earth and rare earth metals, demonstrating the broader applicability of this ligand framework. Additionally, it is shown that three of the rare earth complexes could serve as protonolysis precursors for the synthesis of solvent-free rare earth precursors in an NMR study. As these types of precursors are still limited, the current work of Stevens et al. will surely have an impact on further studies in this field by others.

The overall presentation and scope of the manuscript is good, and I feel it would be suitable for publication in Communications Chemistry after some improvement as per the following more detailed comments.

We sincerely thank the reviewer for their assessment of our work.

Introduction:

- Line 32: there is a word (e.g. "compounds") missing after "The chemistry of AE and RE alkyl"

This has been corrected

- Figure 1: It would be helpful to the reader if underneath each general structure the metals were listed, for which this complex was reported.

This has been done and Figure 1 has been updated accordingly.

- I am missing some explanation on how the metals were chosen in your study, especially for the rare earths?

We thank the reviewer for raising this. We have added motivations for our selection in the Introduction where we explain in more detail the importance of new AE and RE alkyl precursors. We have also added more details in the Results and Discussion section – see discussion after Scheme 1.

Results and Discussion:

- Scheme 1: $M_2(THF)_n$ on the arrows in the left half should be x instead of n

This has been changed.

- Reactivity studies: Please add some discussion on why the study was done only with 1-Yb, 2-Y and 2-La, what about 1-Eu, 2-Pr and 2-Sm?

We thank the reviewer for pointing this out. We chose diamagnetic species that could be monitored *via* NMR. This has been explained more clearly in the text.

- The NMR spectra of the reactivity studies in the SI are too small to tell if the claimed products were actually formed, please add some information on the identification of the reaction products in the main manuscript and annotate the spectra in figures S59, S61, S64 and S67. I also didn't find the 1H NMR spectrum for the reaction of 1-Yb and $HN(SiMe_3)_2$ as well as the phenol, please add these.

We sincerely thank the reviewer for raising this. Following their comments, we had a deeper look at all of the NMR data acquired during these reactivity studies as, in a couple of cases, we noticed some inconsistencies with our original interpretation. Firstly, we are now presenting all the data with more clarity in the SI, showing both full spectra and insets that highlight regions of interests; we have also included the ^1H NMR data for the reactivity of 1-Yb and apologise for the original omissions. In all cases, we were identifying the products using characterisation data already present in the literature – this is now made clear in the text and references have been added (refs. 77, 78 and 79). In the case of the reactivity between 2-RE and $\text{HN}(\text{SiMe}_3)_2$ we noticed that our assignment wasn't entirely correct, complicated also by the close proximity of various signals. Therefore, we decided to make our own references (*i.e.* $[\text{Y}\{\text{N}(\text{SiMe}_3)_2\}_3]$ and $[\text{La}\{\text{N}(\text{SiMe}_3)_2\}_3]$) and use them to analyse the spectra with more confidence. This revealed that in those cases the target complexes hadn't been obtained; a deprotonation reaction is indeed taking place (as suggested by the formation of proligand), but we could not identify which compounds are formed as a result. Accordingly, we have rewritten this section of the manuscript and modified our conclusions to incorporate the different outcomes of these reactions. We apologise for our initial erroneous interpretation of these data and we thank the reviewer for bringing this to our attention.

- NMR discussion: p. 7, l. 169: probably a typo in “ $^{31}\text{P}\{^1\text{H}\}$ ” in the discussion of absence of P-H coupling. Did you measure a ^{31}P NMR spectrum? If so, then please add it to the SI, as currently only the $^{31}\text{P}\{^1\text{H}\}$ spectra are reported.

We thank the reviewer for spotting this. It was indeed a typo and an example of ^{31}P NMR spectrum has been now reported in the SI.

- P. 9, l. 209: please add the theoretical value for μ

This has been done.

- P. 10, l. 246: some Word error, please check

We have not been able to identify this Word error unfortunately.

- P. 13, l. 304: please be more specific what you mean by “more open nature of the NPC-H ligand...”.

We have rephrased this and referred more specifically to the different coordination environments.

- P. 14, l. 315: should be Figure 5, not 6

This has been done.

- P. 16, l. 346: should be Figure 6, not 7

This has been done.

- Figure 6: the wavelength unit should read nm, not cm^{-1} . Can you extend the displayed range in the inset to show the discussed f-d transition at 600 nm?

We thank the reviewer for spotting these mistakes. The label of the x-axis has been updated. With regards to the discussion of the f-d transition, there was a typo in the discussion (now corrected). The transition is at 400 nm, hence the reason for the range displayed in the inset.

- P. 17, l. 365: Please comment on why the study on the electronic structure was restricted to 1-Eu and 2-Sm. I also suggest adding a sentence to the beginning of this paragraph stating that you performed SQUID and EPR experiments as well.

We thank the reviewer for raising this and giving us the opportunity to clarify this aspect of our work. We selected the compounds that were amenable to magnetic characterisation (1-Eu and 2-Sm); all the others are closed shell and therefore not suitable. The only compound that could have been suitable for EPR and SQUID magnetometry is 2-Pr, but we had reservations about its purity, as highlighted from the magnetic susceptibility measurements (Evans method) and in the experimental section (we could never obtain a satisfactory elemental analysis). We have now added a sentence at the start of the paragraph following the reviewer's suggestion.

- Please add the discussion on the results of the electronic structure calculations for 2-Sm to the main manuscript.

We thank the reviewer for raising this point. Given the poor magnetization and susceptibility curves measured for 2-Sm, we decided not to mention the associated CASSCF results as, on their own and without comparison to experiment, they are not informative. We have added a brief sentence in the main text addressing this.

- Figure 7: For which compound is this data?

The data is for 1-Eu. This has now been added to the caption of the figure.

- Are the insights for 1-Eu regarding the mixture of conformers in the EPR relevant for the SQUID measurement?

We thank the reviewer for raising this. Unfortunately they are not informative; we tried simulating the χT and M vs H traces with different D and E values (within the range needed in the EPR) but the changes were minimal.

Experimental section:

- P. 22, l. 502: please specify how high the vacuum was for drying.

This information has been added

- Crystallisation conditions for all complexes: at which temperature were the recrystallizations performed?

This information has been added

- P. 25, l. 574: which colour was the obtained powder?

The powder was very dark in colour, difficult to tell if it was black or maybe a deep blue/purple. We have added this detail to the experimental section.

Supporting information:

- P. 1: please show the ^1H , $^{31}\text{P}\{^1\text{H}\}$ and $^{13}\text{C}\{^1\text{H}\}$ spectra of NPC-H2 as well.

All the spectra are now reported in the SI.

- For the diamagnetic complexes please add integrals to the ^1H NMR spectra

This has been done.

- Fig. S24: Was this spectrum measured on 1-Ca or 1-Ca-THF? The structure shows a THF, but the caption says 1-Ca.

This was for 1-Ca. The structure has been updated.

- Fig. S25: The structure shows only one THF, but the compound is labelled as 1-Sr(THF)₂, please

double check. Is the number of coordinated THF which can be deduced from the ^1H NMR spectrum the same as in the crystal structure?

Coordinated THF molecules are partially lost upon drying the crystals (we used crystalline samples for all of our analyses), which is reflected in the integration of the THF peaks in the ^1H spectra for both 1-Sr and 1-Ba.

- Fig. S47: Please add the peak assignment and the structure.

All pictures have been updated.

- Fig. S49: Please add the peak assignment and the structure.

All pictures have been updated.

- P. 27, top spectrum: The caption is missing.

All pictures have been updated.

- Fig. S58: what are the additional signals observed at ca. 25 ppm starting at 4 hours? Same in Figure S60 and S62

We thank the reviewer for raising this. Following our new analyses of the reactivity studies (see above), we believe these signals could be attributed to heteroleptic methanide complexes. However we do not have enough evidence to unequivocally assign these.

- Fig. S59, S61, S64 and S67: as mentioned above, please assign the signals of the products

This has been done for all reactivity studies.

Reviewer #2 (Remarks to the Author):

The electronic structure and calculations are done for $[\text{Eu}(\text{NPC-H})_3]$? but this complex is not completely characterized.

We would like to point out, respectfully, that the reviewer is incorrect. The compound in question is $[\text{Eu}(\text{NPC-H})_2]$, which has been very thoroughly characterised – see further details below.

No crystal structure of $[\text{Y}\{\text{C}_6\text{H}_4\text{P}(\text{Ph})_2=\text{NnBu}\}_3]$ was obtained and so this structure was not completely characterized either.

We apologise for the confusion, but it is important to clarify that the compound in question is not reported in our work. In fact, in this statement we are referring to a literature compound (ref #112) that could have been used to draw comparisons with compound 2-Y. However, the authors of the work in reference #112 could not obtain a crystal structure of this compound, and we highlighted this as there is a paucity of tris-methanide complexes that we could use for structural comparisons. We have made this more clear in the text to avoid any confusion.

Thus the focus really seems to be the alkaline earth complexes and Sr which are in more detail? **We respectfully disagree with the reviewer. This work presents 4 alkaline earth complexes (+ 3 different solvent adducts and polymorphs, but essentially analogous compounds) and 7 rare earth complexes, so there is a good balance between the two aspects of our work. The alkaline earth complexes are all diamagnetic, hence why their spectroscopic characterisation is discussed in more detail. Nonetheless, there are extensive sections of the paper which discuss the characterisation of all RE complexes, including a dedicated section on the electronic structure characterisation of 1-Eu.**

This work appears to overlap with previous work by some of the same authors, as reported in Polyhedron 2024 (reference 6.)

We respectfully disagree with the reviewer's comment. The only overlap is the use of the potassium transfer reagent developed in reference #6, and at no point we claim that this potassium reagent used in salt metathesis reactions is novel. We have reported 13 new structurally authenticated compounds and none of them are compounds reported in reference #6, nor are the protonolysis reactivity studies or the EPR characterisation of 1-Eu.

Many complexes of beta-diketamates are reported and would make for an interesting comparison.

Although it is undoubtedly true that there are numerous examples of AE and RE beta-diketimate complexes, we respectfully disagree with the reviewer in the assertion that these should be used to draw comparisons for the following reasons: 1) diketimates are not alkyl/methanide ligands (which is the main focus of this work); 2) diketimates form 6-membered chelate rings with the metal centre, whilst our systems form 4-membered chelate rings – so two significantly different coordination modes; 3) diketimate precursors cannot be used as synthetic precursors for deprotonation reactions, at least in the same way as alkyl derivatives. Nonetheless, we have referenced some Yb diketimate complexes in other parts of the paper as these are relevant for comparing the ^{171}Yb NMR data (see other comment below).

This work appears to lack focus or to be incomplete. What trend or utility are the authors trying to demonstrate?

We thank the reviewer for giving us the opportunity to clarify these aspects of our work. We have expanded the discussion on the potential utility of solvent-free alkyl reagents as synthetic precursors, highlighting the lack of precedents for the use of similar methanides as starting materials. Regarding the lack of focus, we apologise if this didn't come out strongly enough in our previous submission. We have now added additional sections to the introduction that put our work in context within modern AE and RE organometallic chemistry research, introducing more clearly the fundamental premise of our work. In particular, we have added this paragraph towards the end of the introduction:

“We were therefore surprised by the fact that the use of the aforementioned methanides in deprotonation reactions had been completely overlooked. We reasoned that a potential deterrent towards the use of these chelating methanides could be identified in the high stability of some of these complexes and also, in the case of RE metals, the lack of homoleptic tris-methanides. Therefore, we were intrigued to explore new methanide systems that could incorporate some of the electronic and steric stabilisation of BIPM-H, and the additional tunability of the asymmetric $\{S=P(Ph)_2CHSiR_3\}^-$ methanide, which could then be employed to stabilise solvent-free AE and RE reagents for deprotonation reactions”

All these additional sections now clearly highlight the great interest of the synthetic community in developing new AE and RE alkyl reagents, which is exactly the focus of our work.

We respectfully disagree with the reviewer when they question the level of completeness of our work, but we appreciate that they probably raised this point based on the perceived lack of focus of our work (which we have now fully addressed). Additionally, we would like to take this opportunity to clarify some important points about the level, quality and completeness of our work. Because we have prepared such a large family of novel complexes, it is expected that these should also be characterised to a very high level – otherwise our work would indeed be incomplete and not suitable for publication. Hence why we produced a significant volume of data to support our findings, using a number of physical characterisation techniques including (when appropriate) advanced spectroscopic and theoretical methods – something that the same reviewer has also noted in their follow-up comment (see below).

We hope that all of these points and additional work address the reviewer's concerns.

While this does appear to be a great deal of work, and I commend them on their detailed characterization, the authors have not done as great of a job supporting their interest in this area nor do they give a compelling argument to its novelty. The paper is a detailed experimental report and does not have an introduction describing other interest in this area or why these compounds are notable.

We thank the reviewer for their appreciation of our experimental work, and we also thank them for raising some concerns about our efforts to showcase the wider interest and novelty of our work. We have now addressed this by significantly expanding the introduction, which now includes more robust reasons for the motivations underpinning our work, also showing the great interest and applicability of alkyl reagents for the development of new synthetic methodologies. We have also highlighted the big gap in the literature with regards to the use of similar class of methanide complexes as synthetic precursors, clarifying their potential advantages with respect to more classic alkyl precursors – see also our response to the previous comment raised by the

reviewer.

This work should include more introduction about other research in the area. There are only 2 references from 2024 in what is arguably a wide field of lanthanide complexes. There is only one reference - by the corresponding author referring to a book chapter reviewing interest in the area.

We thank the reviewer for bringing this to our attention. We have expanded the introduction and the discussion of previous literature, including a significant number of new references and reviews (e.g. Roesky's recent Chem Rev article, and several recent publications from the last 5 years).

There is not good comparisons as to how these structures compare to related complexes of the lanthanides.

We respectfully disagree with the reviewer. In the case of the 1-M family (Sm, Eu, Yb) we have provided extensive comparisons with relevant literature examples. In the case of RE³⁺ compounds (Y, La, Pr, Sm), we have made very clear that there are very few tris-methanide analogues available for meaningful structural comparisons, with the exception of the small family of compounds mentioned in references 108-110. Nonetheless, we have further expanded the discussion to include η^3 -allyl donors (references 111 and 112), and also included a structural comparison of the new complex 1-Sm with compounds in references 25 and 51.

The synthesis is done by protonolysis or salt elimination so not particularly noteworthy. The focus should be on emissions or materials characterization then.

We have indeed focused a lot of our discussion on the spectroscopic and structural characterisation, reactivity, and, whenever possible, magnetic studies.

Its not clear to me what the authors are trying to show with the 1H VT-NMR study of 1-Mg at 25 – 105°C.

We thank the reviewer for the opportunity to clarify this aspect of our work. When analysing the ¹H and ³¹P NMR spectra of 1-Mg collected at room temperature, the compound appears to be impure at a first glance. VT NMR studies in the high temperature regime show that this is not the case, displaying coalescence of all these signals and giving much-simplified spectra. Therefore, this study shows that the multiple signals shown at room temperature arise from different conformations and restricted rotations, rather than impurities. In short, the VT NMR studies are essential to confirm the purity of this compound and provide accurate assignment of the spectra. We have now mentioned this more explicitly in the manuscript by adding the following text:

“We carried out a variable temperature ¹H NMR (VT-NMR) study in d₈-toluene in order to ascertain whether these multiple signals were due to the presence of multiple conformations, rather than impurities. At 105 °C these signals coalesce into a single broad doublet (δ_{H} 1.02 ppm, $^2J_{\text{HP}} = 14.7$ Hz) thus suggesting that there are no impurities and all signals belong to the same species (1-Mg), with restricted rotations or locked conformations overcome in the high temperature regime (Figure S6 and S17-19).”

The Yb NMR data is interesting and unique - but even if no other Yb data for closely related complexes exists - how does this compare to other complexes?

We believe we have already addressed this in the manuscript. The long paragraph on page 9 starting after Table 1 (“The $^{171}\text{Yb}\{^1\text{H}\}$ NMR spectrum of 1-Yb contains a single broad resonance(...)”) provides an extensive analysis of the ^{171}Yb NMR data with comparisons with the literature, comprising 7 references and detailed comparisons with several compounds contained in these manuscripts. Nonetheless, we have now further extended our analysis and included comparisons with seminal work carried out by Lappert and co-workers on ^{171}Yb NMR (references 77 and 87).

The Eu complex seems an outlier as it is not isolated and structurally characterized. This should have been left out or in a separate theoretical chemistry article.

We believe there has been some confusion here. 1-Eu has been isolated and fully characterised via multinuclear NMR, Evans method, CHN, IR, UV-vis, EPR, SQUID magnetometry and single crystal X-ray diffraction (which formed the basis for our attempt to model the magnetic data). Therefore, we respectfully disagree with the reviewer and strongly believe this compound should be included in the manuscript.

Reviewer #3 (Remarks to the Author):

The manuscript describes the synthesis and characterisation of a series of alkaline earth and lanthanide complexes of a silicon- and phosphorus-stabilised carbanion. In general, the work is well done and well described. Some of the compounds are not isolated cleanly, which is a shame, but sufficient evidence of composition is provided. The characterisation methods are very comprehensive, but the data do not always correspond (see below). The authors may wish to consider editing the MS to remove inconsistent calculation data, focusing on the extensive experimental evidence that they have. I recommend acceptance of this article, after the foregoing and the following points have been addressed:

We sincerely thank the reviewer for their positive assessment of our work. With regards to the theoretical calculations and our attempts to model the magnetic data, we believe that our attempts should be presented to the reader in some form. We agree with the reviewer that the focus should be on the experimental evidence, hence why we have kept the discussion of theoretical calculations to a bare minimum. It is important to note that our calculations performed on 1-Eu give a good match with the experimental susceptibility and magnetisation data. Nonetheless, we also clearly state in the manuscript that despite our best efforts the conformational complexity of 1-Eu affects the EPR data in a way that makes the modelling an impossible challenge. All of this despite the use of state-of-the-art and extremely computationally demanding CASSCF-SO calculations, which is an important point that should be communicated to the readership – we have modified some of our discussion (see below) and the conclusions accordingly. We hope we have clarified why we feel the need to include a short analysis of our theoretical work in the manuscript, as our aim is to inform the reader and communicate these important challenges to the community.

Abstract and page 2 line 48: Protonolysis refers to the transfer of a proton from A to B. As written, the correct term for the described reaction is deprotonation.

We thank the reviewer for raising this. The term “protonolysis” has been replaced with “deprotonation” in the text.

Page 3 line 67: “In addition...AlCl₂.” This is not a complete sentence – please correct.

This part of the introduction has been rewritten.

Page 3 line 62 and further: Are the transition and p-block complexes relevant here? Better examples would be the complexes of (Me₃Si)₂C{P(BH₃)Me₂}- with Yb(II) and Sm(II) (Dalton Trans. 2010, 39, 6705), or Mg-Ba (Inorg. Chem. 2007, 46, 4320), or {Ph₂P(BH₃)}₂CH)₂Ca(THF) (Dalton Trans 2009, 2951) – these would appear to be more comparable to the reported compounds.

We sincerely thank the reviewer for highlighting these compounds. These have now been added to the introduction and referenced.

Page 4, line 91: Is it not more likely that the Sm(III) product was produced by exposure of the Sm(II) species to oxygen? Was there any evidence for the formation of Sm metal (dark ppt)? Why was salt metathesis not used to access Sm(II) compounds?

We thank the reviewer for raising this. We agree with the reviewer and cannot exclude the presence of adventitious oxygen. Also, we had originally attempted to isolate the Sm(II) complex via salt metathesis, but our results were inconclusive. However, we decided to try this reaction again and were able to isolate the target complex – synthesis and full characterisation (^1H NMR, Evans method, UV-vis, CHN) now added to the paper.

Page 7: Discussion of the VT NMR spectra of 1-Mg would be aided by including a suitable figure in the text (rather than the SI). The conclusions drawn from these spectra are rather vague – can something more be deduced?

We have now added a new figure in the text (Figure 2) with a portion of the VT NMR study done for 1-Mg (zoomed in the region +/- 0.4 ppm). We have also added the following text to clarify the conclusions obtained from these experiments (see also previous comment in response to reviewer #2):

“We carried out a variable temperature ^1H NMR (VT-NMR) study in d_8 -toluene in order to ascertain whether these multiple signals were due to the presence of multiple conformations, rather than impurities. At 105 °C these signals coalesce into a single broad doublet (δ_{H} 1.02 ppm, $^2J_{\text{HP}} = 14.7$ Hz) thus suggesting that there are no impurities and all signals belong to the same species (1-Mg), with restricted rotations or locked conformations overcome in the high temperature regime (Figure S6 and S17-19).”

Page 8: Why is the ^{171}Yb NMR signal broad? Was a VT experiment attempted?

We thank the reviewer for raising this and for the suggestion. We tentatively ascribe the broadness of the signal in the ^{171}Yb NMR spectrum to unresolved ^{31}P coupling. Unfortunately acquisition of this spectrum requires a very long experiment which would make a VT study impractical. It should also be noted that it's not uncommon observe broad signals in ^{171}Yb NMR (see for example the ^{171}Yb NMR spectrum recently reported by Harder and co-workers in JACS, 2025, DOI: 10.1021/jacs.4c17853).

Page 16, line 351: “This is due to the greater 352 ionicity of these bonds, thereby lowering the energy of the transition into the visible region.” This statement requires more explanation and a reference.

We thank the reviewer for spotting this. This is a sentence that ‘escaped’ from an earlier draft, where we had tried to use DFT calculations to model these transitions (without success). The statement has now been removed.

Page 20: The discussion of the EPR and calculation data is rather poor. The EPR data are experimental and so should be treated as evidence. If the calculations do not correspond with these EPR data, then it must be assumed that the calculations are in error. The explanation of the data as arising from multiple conformers seems unrealistic and should be removed.

We thank the reviewer for raising this. We agree that we should use the experimental data as evidence, hence why we wanted to present our findings despite the challenges in matching some of the data with the calculations. We don't think that it is unrealistic to point at the presence of multiple conformers as a potential reason for the complexity of the EPR data: these conformers are clearly present in the crystallographic model (see positional disorder treatment) and it is reasonable to expect that they would give different crystal field splitting parameters – so the results of these calculations are not surprising. Nonetheless, we do concede that ultimately our

explanations do not shed light on the mismatch between calculations and experimental data, and the reviewer's criticism is perfectly reasonable in this sense. Therefore, we have removed our previous analysis as requested and wrote a new, more generic paragraph, to give the reader the opportunity to understand the challenges of modelling these data without providing any absolute, and potentially misleading, explanations. We hope that the reviewer will find our new approach more acceptable. The new paragraph reads:

“Given the strikingly different parameters set obtained by CASSCF-SO calculations and simulating the EPR spectra, we turned to the molecular geometry looking for an explanation. We note that the presence of extensive crystallographic positional disorder in 1-Eu and the occurrence of different conformers could be responsible for the difficulty in predicting the correct D and E values. We have tried to address this by performing CASSCF-SO calculations using different geometric models (Table S10), but to no avail. Therefore, at present, we are not able to produce a theoretical model that captures the experimental magnetic data of 1-Eu in their entirety.”

Reviewer #4 (Remarks to the Author):

The crystallography has been done well with details of the refinements included in most of the CIFs. The answers to the ALERT A and B's are all reasonable. The crystal structure all support the finding of the authors.

Correction:

Page 10 2.2 Xray crystallography, first line:

....all Error! Reference source not found.

This has been fixed.

Page 19

Figure 10 is mentioned a couple of times but there is no figure 10. I think it should be figure 9 as there is a figure 9re:.

This has been corrected.

SI says: "Crystals were examined using a Bruker D8 Quest diffractometer with a Photon III detector" However, CIF info does not match this in: 1-Ca-THF, 1-Mg-C2/c, 1-Sr, 1-Sr-THF, 2-Yb please correct either the SI or the CIF's. 2-Y appears to have been collect on a Rigaku instrument again correct the SI or CIF.

We thank the reviewer for spotting these. The inconsistencies were due to how Olex2 was used to compile the cif files. All the data have been collected using a Bruker D8 Quest with Photon III; this is also the case for 2-Y, but the data in that case was reduced using CrysAlisPro, hence the confusion. All the information in the cif files has been updated and the information about CrysAlisPro has also been added to the SI.

2-Yb the Centre of Gravity not Within Unit Cell please move the molecule into the unit cell.

This has been fixed.

Comments and question:

Why does CHECKCIF report large numbers of reflections missing such as 1_Ca where 319 reflections (1-Ca-THF) 146, 1-Sr 201, 2-Sm 128 and 2-Y 146) are reported missing. Is this a result of the data collection strategy or processing issues. This is much high than I have seen before.

We thank the reviewer for raising this. In these cases, the missing reflections arise from data processing - this happens automatically when scaling integrated data using Apex4 when dealing with multidomain systems, unless a loose threshold is applied. In all these cases, the other domains could not be isolated an identified so we were not able to apply a rigorous twinning treatment, with the exception of 2-Y which could be treated as a 2-component system.

Were global RIGU comments really needed to get a good refinement in several of the refinements?

We thank the reviewer for raising this. We applied RIGU to aid refinement whenever needed, which often arose from the presence of positional disorder which could not be modelled – this is pervasive in these systems due to the equivalence of the methanide carbon and imido nitrogen of the NPC-H ligand. Nonetheless, we have removed RIGU from the structure 1-Ba as we realised

there was very little improvement when using this restraint.

1-Eu why did you not just use displacement restraints to fix the ALERT B, you seem happy to use them in the other refinements.

We thank the reviewer for pointing this out. We did indeed try to apply restraints (SIMU and RIGU) but these did not lead to an improved model, so we decided to remove them.

1-Ba it would have been more consistent to ride the hydrogen displacement parameter like it the other refinements rather than leaving them to freely refine.

All the hydrogen atoms in the structures have now been refined using the riding model.

Reviewer #1:

The authors have done a good job at addressing my remarks and questions and I have only one small remaining comment: In Scheme 1 in the description underneath 1-M there's still one "n =..." which should read "x = ...". Other than that I'm happy to recommend the manuscript to be accepted for publication.

We sincerely thank the reviewer for their time and for the positive assessment of our work. Scheme 1 has been updated accordingly.

Reviewer #2:

The authors have put in considerable effort to clarify their project details. This is most notable in the improved explanations and the additional references incorporated in the introduction and the description.

The corrections to Figures 1, 6, and 7 have helped improve the text. The addition of the 31P NMR to the SI was also informative.

I don't think it's necessary to include the symmetry operations in the figure caption for Figure 7.

The comparisons to references 116 and 117 are helpful.

Overall I think the authors have done a great deal of additional work and have satisfactorily addressed the reviewers comments.

We sincerely thank the reviewer for their time and the positive assessment of our work. We have decided to keep the symmetry operations to avoid any confusion in case a reader decided to look at individual bond distances and angles.

Reviewer #3:

I am happy that the extensive suggestions made by the reviewers have been adequately addressed and that this manuscript may now be accepted for publication.

We sincerely thank the reviewer for their time and for the positive assessment of our work.

Reviewer #4:

I am happy with the corrections and explanations to my comments. The Crystallography is to a high enough standard to support the conclusion in the manuscript.

We sincerely thank the reviewer for their time and for the positive assessment of our work.